# Protect or prevent? A practicable framework for the dilemmas of COVID-19 vaccine prioritization

**Raghu Arghal**[1]*, **Harvey Rubin**[2], **Shirin Saeedi Bidokhti**[1], **Saswati Sarkar**[1]

**1** Department of Electrical and Systems Engineering, University of Pennsylvania, Philadelphia, PA, United States of America, **2** Division of Infectious Diseases, Department of Medicine, University of Pennsylvania School of Medicine, Philadelphia, PA, United States of America

* rarghal@seas.upenn.edu

**Data Availability Statement:** All data and code are available on Github at https://github.com/raarghal/ProtectorPrevent.

**Funding:** The author(s) received no specific funding for this work.

## Abstract

Determining COVID-19 vaccination strategies presents many challenges in light of limited vaccination capacity and the heterogeneity of affected communities. Who should be prioritized for early vaccination when different groups manifest different levels of risks and contact rates? Answering such questions often becomes computationally intractable given that network size can exceed millions. We obtain a framework to compute the optimal vaccination strategy within seconds to minutes from among all strategies, including highly dynamic ones that adjust vaccine allocation as often as required, and even with modest computation resources. We then determine the optimal strategy for a large range of parameter values representative of various US states, countries, and case studies including retirement homes and prisons. The optimal is almost always one of a few candidate strategies, and, even when not, the suboptimality of the best among these candidates is minimal. Further, we find that many commonly deployed vaccination strategies, such as vaccinating the high risk group first, or administering second doses without delay, can often incur higher death rates, hospitalizations, and symptomatic infection counts. Our framework can be easily adapted to future variants or pandemics through appropriate choice of the compartments of the disease and parameters.

## 1 Introduction

Since its beginning in December 2019, the COVID-19 pandemic has resulted in nearly 750 million infections and over 6 million deaths as of April 2023 [1]. Vaccines have proven to be the most effective countermeasure to the pandemic by limiting transmissions and protecting especially vulnerable populations [2]. During its early stages, the vaccination drive was heavily capacity constrained with demand far outstripping supply and administration capability—a challenge that continues to plague Low- and Middle-Income Countries (LMICs) [3]. This is bound to be the case for vaccines developed for every infectious disease.

The target population for COVID-19 is naturally heterogeneous with different individuals exhibiting different social contact patterns and different risks for developing a serious form of

**Competing interests:** The authors have declared that no competing interests exist.

the disease and suffering hospitalization or death. Thus, under capacity constraints, governments and public health organizations must make the critical choice of whom to vaccinate first: 1) those who are at high risk of hospitalization or death 2) those who are likely to transmit the disease most, 3) a combination of the first and second set. The first group, referred to as the "high risk group", is comprised of the elderly and the immunocompromised; vaccinating them protects them by significantly reducing the risk of severe symptoms, hospitalization, and death. The second, the "high contact group", are those who manifest high contact rates, have large and dynamic contact sets, perhaps because of their professions (e.g. shopkeepers, bank tellers, receptionists, drivers of taxis, shared rides, waiters, bartenders, doormen), and may spread the disease the most; thus, vaccinating them aims to prevent them from infecting others, significantly reducing the reach of the disease in the populace, including in the high risk group. For COVID-19, most public health bodies prioritized the vaccination of the high risk group over the high contact group, or rather over a large segment of the high contact group, adopting the protection route [4]. The only segment of the high contact group that was prioritized over the high risk group (e.g. the elderly and immunocompromised), were health care workers, but they form a negligible segment of the high contact group. But was the choice to focus on protection optimal even if we consider the objective of minimizing only the death count? What if the prevention route minimizes the death count instead, by vaccinating first those who constitute the most effective vector of spread, which in turn reduces the spread of the disease to the most vulnerable? In particular, if a small fraction of the populace have much higher contact rates than the rest, then vaccinating them may inhibit in a short time the pathway of the disease to the most vulnerable.

Next, while some vaccines have one dose, others need two doses [5]. For the recipients who opted for the vaccines that needed two doses, the public health bodies, at least in the US, scheduled the second dose right after the mandated minimum time between the two doses, while others await their first doses. But do public health metrics improve if first everyone is administered the first dose and then the second doses are scheduled for those who need them? Or is the optimal strategy a hybrid between the two extremes, i.e. delay the second dose by a certain amount to administer the first dose sooner to more individuals; the delay may be different for different contact and risk profiles as well. To answer these questions, we need a systematic methodology to determine the optimal vaccination strategy, for attaining well-defined public health objectives, considering both single dose and two-dose vaccines. Identifying the optimal vaccination strategy is imperative also because it provides a valuable benchmark for evaluation of any proposed vaccination policy. Comparing a proposed policy with the optimal informs by how much the former can be outperformed through a smarter design.

The challenge in determining the optimal vaccination strategy for COVID-19 is multi-fold. First, COVID-19 manifests some fundamentally different characteristics in susceptibility and fatality rates as compared to other contagious diseases such as various strains of influenza, rabies, which have been known for a while, and, more recently discovered ones such as Zika and Ebola [6]. Specifically, vulnerability to COVID-19 has shown to be much more lopsided with respect to age (presenting greater risk to the elderly) as well as other underlying health conditions such as diabetes, obesity, or hypertension [7]. For example, patients above 65 years of age and patients with diabetes with complications experience increases in mortality risk of over 500% and 30%, respectively, when compared to healthy individuals aged 18 to 39 [8]. Complications here refer to conditions (e.g. heart conditions, kidney damage, and diabetic neuropathy) which are caused or exacerbated by diabetes [8]. In fact, 81% of US COVID deaths in the first year of the pandemic were individuals over 65 [9]. Thus, given the nature of transmission and vulnerability, the extensive body of research that exists for vaccination for the other diseases mentioned above, could justifiably assume either a fully homogeneous target

populace or limit heterogeneity to age-based compartments or geography [10–12]. For example, studies of rabies have rightly focused primarily on geographic movement of animal populations which are common hosts (eg. [12]). The age or geography based compartments were often considered to differ in contact rates, but not in risks, even when they consider heterogeneity. Thus, the literature on vaccinations for other diseases do not incorporate the heterogeneity in *both* contact rates and risk level which constitutes a defining characteristic for COVID-19. Yet, incorporating this multi-dimensional heterogeneity by considering each individual as a separate entity usually leads us to optimizations of an inordinate size, which are computationally intractable because of the curse of dimensionality and the scale of spread of COVID-19 [13]. On the other hand, homogeneous abstractions, though computationally tractable, lose the essential characteristics of the spread and impact of COVID-19. Even the scale of spread of some of the previous diseases such as Zika or Ebola have been far more limited than COVID-19, with Ebola outbreaks being largely confined to West Africa [14], while COVID-19 afflicted all the continents. The much larger target populace amplifies the computation challenges that arise in determining the right prioritization based on individual characteristics such as contact rates and risk factors.

We redress the above challenges as follows. We model the multi-dimensional heterogeneity for COVID-19 by grouping different sections of the populace in accordance with their risk factors and contact rates. Individuals in each group are assumed to be statistically identical in terms of risk factors and contact rates among each other and across groups, but those in different groups can have different contact rates and risk factors. Contact rates can vary significantly depending on profession and age. For example, retail salespersons have a reported mean number of contacts per day of 62 while the same figure for computer programmers was just 15 [15, 16]. Contact rates within groups can also be different from those between groups. As another example, contact rates among people under 65 in US have been reported to be above 8 contacts per day, and contact rates between people below 65 and people over 65 have been reported to be 0.6 contacts per day or lower [17]. The contact rates between groups depend on pairs in question. This group-based modeling provides a tradeoff between retaining analytical and numerical tractability and capturing the inherent heterogeneity. The number of groups and the specific criteria for the classification is a design choice. Vaccination in different groups however remains coupled due to capacity constraints on overall rate of vaccination across all groups together. We subsequently formulate the determination of the optimal vaccination strategy that optimizes public health metrics of choice (eg, death, hospitalization, symptomatic counts) by allocating different vaccine capacities to different groups at different times subject to the limit on the number of doses that can be administered (i.e. the vaccination capacity constraint).

There is a growing literature attempting to provide guidance on COVID-19 vaccine prioritization. Most papers compare a set of pre-identified policies using simulations e.g. [18–23]. The most common among these policies prioritize based on age (e.g. [18]), others based on geographic locations etc [19–21]. Among these, a small set focus on LMIC settings (e.g. [22]) with the large majority confined to high income countries [23]. Incidentally, among prior pandemics, influenza may be the closest to COVID-19, and most vaccination strategies for influenza fall into similar categories, relying either on simulation of very few strategies (e.g. [24, 25]) or computationally costly optimization with limits on the feasible set of policies (e.g. [26, 27]). Such comparisons and evaluations often provide valuable insights, but the overall approach suffers from some fundamental limitations. First, in absence of the knowledge of the optimal vaccination strategy for attaining well-defined public health objectives, one does not know whether and by how much a future strategy, or even all past strategies, outperform the proposed strategies. Second, simulations require a large number of iterations to provide

reliable performance estimates and each such iteration consumes significant time for even moderate size populaces, and the computation time and processor memory requirement for each iteration grows at least polynomially with the size of the populace. As a result, the works in this category which consider populations of the size that pandemic spread affects rely heavily on access to powerful computing resources like high performance computing clusters (e.g. [28, 29]). These simulations cannot therefore be conducted in settings which have limited computational resources such as local public health bodies in small towns of the US and throughout LMICs, which prevents the choice between strategies from being fine-tuned to parameters that evolve with time and can only be locally estimated. Barring few exceptions, most works consider single dose vaccines, though several COVID-19 vaccines require two doses. For example [30, 31] consider two dose vaccines, but both of these prioritize based only on age, as per a predetermined order, and choose between few predetermined delays for the second dose. The joint optimization of group prioritization and second dose delay has remained open even in the category that compares a set of predetermined strategies.

Another computational approach chooses the optimal vaccination strategy for COVID-19 among the policies that do not vary the vaccination rates to groups over time (i.e. among static policies) (e.g. [32]), or among those that vary allocations over large predetermined time intervals (e.g. [6]). The limitation of these approaches is that they *a priori* rule out highly-dynamic vaccine allocations without considering if these can attain substantially better values of public health metrics. Besides, the appropriate time scale of optimization can not be determined *a priori* and universally, as different public health domains have different inherent flexibilities. The available approaches that optimize over predetermined fixed intervals can not easily generalize to different, specifically higher, numbers of intervals, i.e. they do not scale. For example Buckner et. al. deploy a two-step genetic algorithm with simulated annealing [6]. The computation time of genetic algorithms increases exponentially in the number of optimization variables. The number of optimization variables for Buckner et al. linearly increases in the number of intervals. Thus, the computation time significantly increases if the number of intervals increases (refer to S5 Appendix for more details). Again, these approaches may be prohibitively difficult in areas lacking advanced computing or the necessary expertise. All of the above only consider single dose vaccinations, and the generalization to two doses is not direct and is expected to increase the computation time even further and significantly so.

We adopt a different design approach altogether. We do not restrict to any preselected set of vaccination strategies, or any predetermined decision interval. We instead consider the problem in its most general form, optimizing over all potential vaccination strategies, including arbitrary functions of time which can vary over any time scale and arbitrary variations across the groups, towards the desired public health metric subject to the vaccination capacity constraint. We obtain a flexible framework that can accommodate different number of doses and different public health metrics (e.g. death, hospitalization, symptomatic counts), and tailor the vaccination strategy to any contact and risk heterogeneity of the target populace, population demography, disease parameters, and vaccination capacity constraints. We accomplish this in three broad steps: (1) capturing the heterogeneity of the populace in both contact rates and risk factors through a novel partition of the populace into three groups: high contact, high risk, and baseline (Section 2) (2) modeling the spread of the contagion and the progression of the stages of the disease within and across the groups, as a system of a small number of ordinary differential equations (ODEs) in a small number of variables, regardless of the size of the populace (Section 2) (3) posing the choice of the optimal capacity constrained vaccination strategy as an optimal control problem with the ODEs providing the state trajectories (Section 2). Optimal control has been deployed in designing dynamic vaccination prioritizations for several other contagious diseases (e.g. [33, 34]), but the deployment for COVID-19 will

fundamentally differ on account of its distinctive characteristics. Overall, very few papers have applied optimal control to design of COVID-19 vaccine strategies [21, 35, 36]. [35, 36] both deploy optimal control to determine the optimal budget allocation between several different types of mitigation strategies for COVID-19 (e.g. social distancing, vaccination, testing) assuming that the target populace is fully homogeneous. [21] focuses on allocation of available vaccines among large sub-regions (e.g. provinces of a nation, with Italy as their case study) of a geographic region as to minimize cumulative infections. They consider the population in each sub-region to be homogeneous in that every individual has the same risk factor and contact rate therein. In the above respective scenarios, the fully homogeneous abstraction, or the homogeneity assumptions within each sub-region rules out design of vaccine prioritization among individuals based on their contact rates and risk factors. Thus, the problems they seek to solve are complementary to those in this paper. Also, note that in practice different individuals within large or even small sub-regions (e.g. neighborhoods) have widely diverging contact rates (e.g. due to their professions) and risk factors (e.g. due to their age and underlying health conditions). Thus, even the vaccine rollout processes deployed in practice have enacted different priorities among different individuals living even in proximity [4]. Thus, our consideration of heterogeneity captures a crucial element of reality.

We compute the optimal vaccination strategy for 911,250 instances of realistic parameters (see S4 Appendix for parameter selection), spanning different variants, disease parameters and population demographics of 139 countries and all states of US as well as specific case studies such as retirement homes, prisons, LMICs. The optimal solution, or a close approximation thereof, turns out to be easily deployable, and can mostly be computed within seconds regardless of the population size using standard numerical toolboxes and modest computation resources, which are usually accessible in local public health offices throughout US and in most LMICs. Owing to this computational tractability, we are able to evaluate the impact on the solution of 1) a range of disease and population parameters that arise in practice including in divergent case studies of interest, 2) restrictions on decision intervals, and 3) error in estimating the parameters. Finally, we show how our framework can be used by public health authorities in designing vaccine rollout strategies for future pandemics (Section 4).

We summarize some specific findings from our work. We first list those that are particularly relevant for practitioners.

- For one dose vaccines, we could narrow down the set of optimal strategies to only two candidates: one which first vaccinates the high contact group, then the high risk group and finally the baseline group, the other reverses the order between the high contact and high risk groups (Section 3.2). One or other of these two policies optimizes important public health metrics among all policies for the bulk of the parameters considered, and nearly minimizes in the limited number of remaining instances. This is a significant reduction starting from the set of all possible vaccination strategies. Further, both candidate policies are easy to deploy. while the commonly recommended high risk prioritization was the optimal policy in some instances, in many realistic cases (e.g. nursing homes and LMICs), the high contact priority policy has a lower death count than the high risk priority policy which is currently the most widely deployed policy; for some realistic cases the high risk priority policy has higher death count than even a policy that accords uniform priority to all individuals i.e. randomly selects among them (Sections 3.3 and 3.4). This is somewhat counterintuitive because the high risk priority policy prioritizes the vaccination of those at greatest risk to die once infected, but has been explained through the insights the computations provide.

- For two dose vaccines, we find that the set of optimal vaccination strategies can be narrowed to three easy to deploy candidates, which vary in the order of selection of groups for

vaccination and on whether the second dose is delayed until everyone receives the first dose or each group gets the two doses in succession while others wait for the first dose (Section 3.8). It is again not obvious *a priori* that the candidate set for two doses will increase only by a small number compared to that for one dose given the several additional decision variables in the former, including one decision variable, namely the delay for the second dose, assuming uncountably infinite number of possible values being a real number.

These two findings imply that the optimal vaccination strategy, or a close approximation thereof, can be determined by comparing a small set (i.e. either two or three) of vaccination strategies, for both one and two dose vaccines. The comparisons can again be done in a few seconds using our framework even when the computation resources are modest. This facilitates the universal deployment of the framework by enabling fine tuning the choice between the few candidate policies to specific parameters that can only be identified locally.

We now highlight the findings that have significance from a methodological view.

- We show that despite segmenting the population into only three groups (baseline, high risk, and high contact), when the parameters are chosen correctly, our model is able to accurately predict both the spread of infection and death count of COVID-19 as observed over the course of the pandemic for all US states and 139 other countries including most LMICs (Section 3.1). Thus, our model is universally applicable. We also demonstrate that any decrease in the number of groups substantially undermines the accuracy of prediction. In this sense, our model has just enough complexity to characterize the state dynamics of COVID-19 so as to match actual data of spread and inflicted mortality.

- We demonstrate the use of optimal control as an alternative to simulation or other, more computationally expensive optimization techniques for determining vaccine prioritization strategies for pandemics in which the target populace manifests heterogeneity in both contact rates and risk factors.

Overall, our framework for the determination of the vaccination prioritization is practicable in (1) the ease of deployment of all the resulting candidate solutions, (2) the fast computation of the optimal solution even when the computation resources are modest as in local public health units in US and LMICs which allows such units to fine tune the solution they deploy in accordance with the local pandemic parameters, and (3) the ability of the underlying model to capture key public health metrics almost universally.

## 2 Methods

COVID-19 manifests heterogeneity both in the evolution of the disease in individuals and in the manner of the spread. Different sections of the populace constitute the most potent vector of spread while others have high risk for developing a severe form of disease once infected. We capture this heterogeneity through a simple classification of the populace. We divide the populace of N people into three groups: 1) baseline (X) 2) high risk (Y), and 3) high contact (Z). We describe the rationale and method underlying this classification. The *high risk* are those who are old or have underlying health conditions, which exacerbate the risk of developing severe forms of the disease, namely symptoms, hospitalization and death, once infected, e.g. retirees [7]. The *high contact* group are those who have a large set of potential contacts spanning entire neighborhoods or even cities, but establish actual contact with any individual in their set of potential contacts infrequently. The *baseline* is the rest of the populace, i.e. those who are neither high contact nor high risk.

The high contact group includes shopkeepers, bank tellers, receptionists at hotels, restaurants, waiters at restaurants, doctors, nurses, drivers of buses, cabs and shared ride, etc. For example the set of potential contacts of a shopkeeper or a driver includes entire neighborhoods or cities, but a shopkeeper or a driver meets any given member in their neighborhood or city infrequently. This group has overall high rates of contact with both of the other groups. Given their large and dynamic contact sets, individuals in this group are the most likely to imbibe an infectious disease early on and pass the same to a large number. Thus, this group constitutes the most potent vector of spread. The individuals who are not high contact, have small sets of "regular" contacts, (e.g. family members, colleagues, friends), with whom they connect frequently, and outside this set infrequently connect with members of the high contact group. The baseline group includes but is not limited to workers of the tech sector, financial sector, manufacturing sector, homemakers, and scientists. Note that a tech worker or a scientist would regularly meet their friends, family, colleagues—all of whom constitute a small set; and infrequently meet shopkeepers, bank tellers, waiters, drivers in their neighborhood or city. Thus, members in this group largely, albeit indirectly, connect through those of the high contact group.

Considering publicly available data for the infection and death counts as a function of time in all states of US and 139 countries, we later show that the decomposition of the populace in these three groups helps capture the temporal variation of infection and death counts observed in reality (Section 3.1). We also show, using the above mentioned data, that decomposition into fewer groups, namely one (i.e. entire population is one homogeneous group) or two (separating individuals based only on risk factor or only on contact patterns) can not capture the temporal variation of both infection and death counts (Section 3.1). Thus, our classification is the simplest possible for capturing the heterogeneity in the contact and risk patterns of the populace which determine the spread and evolution of COVID-19. And, the classification based on contact patterns and risk factors allows us to capture the tension between focusing vaccination on high risk or high contact groups.

To describe state transitions modeling the contraction and evolution of COVID-19 in individuals, we employ an expanded compartmental model that accounts for the disease states associated with COVID-19 as well as our three group partition. We imbue each group with a compartmental model of the disease progression which includes 10 states: susceptible (S), exposed (E), pre-symptomatic (P), asymptomatic (A), early stage infected (I), late stage infection (L), hospitalized (H), recovered (R), vaccinated (V), and dead (D). In general, we refer to the fraction of individuals in a disease state with the group as a subscript and indexed by time e.g. $S_X(t) = \frac{N_X(t)}{N}$ where $N_X(t)$ is the number of susceptible, type $X$ individuals at time $t$ and $N$ is the the total number of individuals.

For ease of exposition, we first present a simplified model in which vaccines confer perfect immunity to infection (see Fig 1), an assumption relaxed in the model we actually use for all our numerical results (refer to details of this model in S2 Appendix).

The vaccine policy is specified as the fraction of the susceptibles of each group who will receive the vaccine at a given time instant $t$ ($u_X(t), u_Y(t), u_Z(t)$). Thus, $0 \leq u_i(t) \leq 1$ at any given time $t$. We are therefore assuming that only individuals who do not have active infection, ie, susceptibles are vaccinated. This assumption is in line with CDC recommendations which state that vaccine should be delayed at least until symptoms and isolation criterion subside and ideally until 3 months from the onset of symptoms or positive testing [38]. Furthermore, when vaccine capacity is severely limited (as is bound to be the case early on in vaccine availability or in low resource settings), it is especially important that only susceptible individuals who can reap full benefit receive vaccines [18]. Note that symptomatics can be easily excluded from

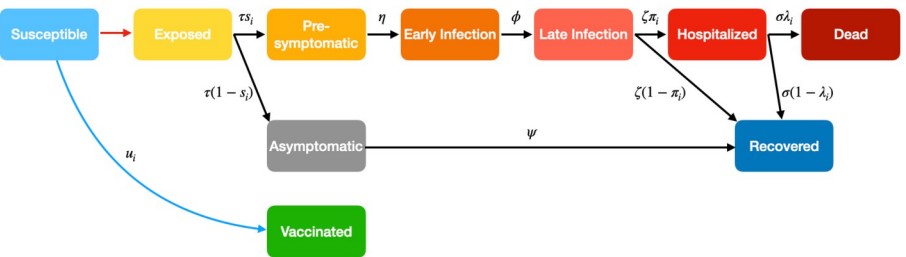

**(a)** Simplified COVID state diagram

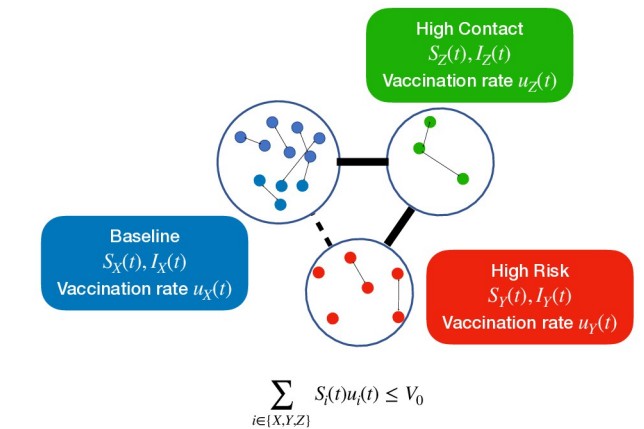

$$\sum_{i\in\{X,Y,Z\}} S_i(t)u_i(t) \leq V_0$$

**(b)** Contact and vaccination model

**Fig 1.** In Fig 1a we present a simplified state diagram in the case where vaccinated individuals and recovered individuals cannot become infected. Note that these assumptions are relaxed for our numerical computations (see S2 Appendix for further detail). Red arrow indicates exposure to a contagious individual, black arrows denote natural disease progression, and blue arrow denotes vaccination. Each transition (apart from exposure which depends on infected population) is labeled with the associated rate. Note that probability of becoming symptomatic ($s_i$), hospitalized ($\pi_i$), and dying ($\lambda_i$) are group-dependent as denoted by subscript $i$. Here $\tau$, $\eta$, $\phi$, $\psi$, $\zeta$, and $\sigma$ are the reciprocals of the expected durations of the exposed, pre-symptomatic, early infection, asymptomatic, late infection, and hospitalized states, respectively. Refer to S1 Appendix for notation tables. Fig 1b depicts our three group model which dictates how susceptible and infectious individuals of different groups come in contact to spread the disease and how vaccines are allocated. Here $I$ denotes a group's vector of all infectious disease states. Note that as vaccination rate $u_i(t)$ for a group increases the fraction of individuals that transitions to the path that takes them to hospitalization or dead states decreases. Given the respective vaccination rates and number of susceptibles, the vaccination capacity constraint is expressed as $\sum_{i\in\{X,Y,Z\}} S_i(t)u_i(t) \leq V_0$. **(a)** Simplified COVID state diagram **(b)** Contact and vaccination model.

vaccination based on manifestation of symptoms. The exposed, asymptomatics and presymptomatics can also be excluded if all individuals are tested before they are vaccinated. Usually by the time vaccines are available for any infectious disease, tests for the same are widely available and are also cheap, thus testing before vaccination can be easily realized. But if vaccination can not be preceded with tests, then the exposed, presymptomatic, and asymptomatic individuals will also receive vaccine doses, we show how our framework can be adapted to this setting in S2 Appendix.

Note that *all* vaccination policies can be represented through appropriate choice of the functions $u_i(\cdot)$. For example, one can consider policies which sequentially vaccinate groups, e.g. $u_i(t)$ is as high as possible for a certain group initially, while $u_j(t) = 0$ for other groups in the same period, then the focus changes to $u_j(t)$ for another group, etc. Or, $0 < u_i(t) < 1$ for all groups in a certain interval, which represents "mixed" policies which split vaccine capacity

between multiple groups. In fact, $u_X(t)$, $u_Y(t)$, $u_Z(t)$ are allowed to be arbitrary functions of time, Thus, highly dynamic, complex policies are included among the vaccination strategies we consider.

We consider that the vaccination capacity is limited, i.e. $V_0$ fraction of the population ($NV_0$ individuals) can receive vaccines on any given day. Thus, $\sum_{i \in \{X,Y,Z\}} u_i(t)S_i(t) \leq V_0$ for each $t$. This vaccination capacity constraint is motivated by staffing and infrastructure limitations that arise in particular for emerging infectious diseases. Specifically the health care workers who administer the vaccines are limited. Also, only a certain number of doses can be stored on any given day in a health care unit due to limitations on cold chain storage space [39]. Thus, only a certain maximum number of doses can be administered on any given day leading to the hard constraint under consideration [39].

The dynamics of the system can be captured by a system of ODEs provided in Fig 2. While the set of ODEs described is deterministic, the state transitions are usually stochastic. It can be shown, however, that, under certain mild regularity assumptions, the fraction of individuals in any given state and group in the stochastic system converges to the solution of the ODEs at each time $t$ in the asymptotic limit that the size of the population increases to infinity ([40]; Supporting Information C, [37]). The assumption is that the duration of each state for every individual is exponentially distributed; under this assumption the distribution of the number of individuals across the states and groups constitutes a continuous time Markov chain. Thus, under this assumption, the solution of the ODEs becomes a more and more accurate approximation of the temporal evolution of the actual state distribution as the population size increases to infinity. This constitutes an important computational strength as the system of ODEs can be readily solved regardless of the population size, while the time required to compute the probability of different population distributions across states and groups from the continuous time Markov chain representation increases exponentially in the population size.

$$\dot{S}_i(t) = {\color{red}-S_i(t)u_i(t)} - S_i(t)\sum_{j \in \{X,Y,Z\}} \beta ij[P_j(t)\rho + A_j(t)mu + I_j(t)\omega] \quad (1)$$

$$\dot{E}_i(t) = -E_i(t)\tau + S_i(t)\sum_{j \in \{X,Y,Z\}} \beta ij[P_j(t)\rho + A_j(t)mu + I_j(t)\omega] \quad (2)$$

$$\dot{P}_i(t) = E_i(t)\tau s_i - P_i(t)\eta \quad (3)$$

$$\dot{A}_i(t) = E_i(t)\tau(1 - s_i) - A_i(t)\psi \quad (4)$$

$$\dot{I}_i(t) = P_i(t)\eta - I_i(t)\phi \quad (5)$$

$$\dot{L}_i(t) = I_i(t)\phi - L_i(t)\zeta \quad (6)$$

$$\dot{H}_i(t) = L_i(t)\zeta\pi_i - H_i(t)\sigma \quad (7)$$

$$\dot{V}_i(t) = {\color{blue}S_i(t)u_i(t)} \quad (8)$$

$$\dot{R}_i(t) = A_i(t)\psi + L_i(t)\zeta(1 - \pi_i) + H_i(t)\sigma(1 - \lambda_i) \quad (9)$$

$$\dot{D}_i(t) = H_i(t)\sigma\lambda_i \quad (10)$$

$$i \in \{X, Y, Z\}$$

**Fig 2. Here we present a set of ordinary differential equations (ODEs) which captures the dynamics of the states shown in Fig 1.** All notation and associated parameters are defined and presented in full detail in S1 Appendix. These ODEs have been obtained through an adaptation of the metapopulation model of epidemiological differential equations [37]. Each equation governs the rate of change of the fraction of individuals of a specific group and disease state. The equations are composed of both quadratic and linear terms. The quadratic terms (terms in red) correspond to the transmission of the virus, which involves interaction between two individuals, one contagious and another susceptible. The number of such interactions per unit time is linearly proportional to the number of such pairs, which is in turn linearly proportional to the product of the fraction of susceptibles and contagious individuals in the respective groups. We assume that individuals are contagious in the asymptomatic, presymptomatic, and early infection states after which, they are either recovered, quarantined, or hospitalized and unable to spread the disease further. The proportionality constant is the disease spread rate. Linear terms correspond to either (1) the progression of an individual through disease states (terms in black) or (2) the vaccination of susceptible individuals (terms in blue). The proportionality constants for (1) are the transition rates in and out of the states which are different for different groups. The proportionality constants for (2) are the vaccination rates which are functions of time and group. This system of ODEs consists of only 30 ODEs involving 30 variables, regardless of the population size. Note that this represents the simplified setting in which vaccinated and recovered individuals cannot become infected (corresponding to Fig 1).

Thus, this latter computation is intractable for all practical purposes for populations of even moderate size.

The state dynamics of the three groups are connected through (1) the infection across groups and (2) the dependence between the vaccination rate $\{u_i(\cdot)\}$ allocated to each group at time $t$. The dependence arises because of the vaccination capacity constraint. If the capacity allocated to one group is high at any given time, those allocated to the other groups must be low because of this capacity constraint.

We now seek to optimize public health metrics of our choice through a judicious selection of the vaccination strategy, namely the $\{u_i(t)\}$ for $i = X, Y, Z, t \in [0, T]$. Unless otherwise specified, we focus on minimizing the overall death count by the end of the time horizon under consideration, i.e. $N(D_X(T) + D_Y(T) + D_Z(T))$ (this is also the cumulative death count in $[0, T]$). We later describe how our framework can be adapted to minimize other public health metrics such as symptomatic counts, hospitalization counts, or socioeconomic costs related to the pandemic (S2 Appendix).

The optimal vaccination strategy is the one that minimizes the overall death count among *all* vaccination strategies that satisfy the vaccination capacity constraint. Since the available choices include arbitrary functions of time, the optimal vaccination strategy can not be obtained by standard optimization, but needs to be characterized by solving an optimal control formulation, which we will shortly describe.

The first question however is if the vaccination strategy that minimizes the cumulative death count needs to be obtained by solving any optimization or optimal control problem at all. Wouldn't vaccinating all individuals in the high risk group first accomplish this goal always? To see why not consider the case that the high risk and high contact groups are respectively large and small in size, and the contact rates between the high contact group and other groups is much higher than the contact rates between other groups and within other groups. The initially infected individuals are all in the baseline group. Then, if the high risk group is vaccinated first, it will take some time to complete the vaccination strategy given the group's size. During this time the infection may be transmitted from the baseline group to the high risk group through the high contact group. Once the infection reaches the high risk group it kills many individuals in this group before they can be vaccinated. Instead if the high contact group would be vaccinated first, its vaccination would be completed in a short time because of its small size and the virus may not be able to reach the high risk group in this time. Once the high contact group is vaccinated in its entirety, the virus can not reach the high risk group as it lacks a path to it from the baseline group in which the infection had originated. Thus, the death count in the high risk group is low. The mortality risk is low in other groups, so very few individuals therein die even if they are infected. Thus, the overall death count is low. In other words, vaccinating the high contact group before the high risk group may in fact reduce the death count as compared to when this order is reversed. Thus, the optimal vaccination strategy is not *a priori* obvious. We therefore proceed to formulate the optimal control problem:

$$
\begin{aligned}
\text{minimize} \quad & \sum_{i \in \{X,Y,Z\}} N \cdot D_i(T) \\
\text{subject to} \quad & x \in \mathcal{S} \\
& x(0) = x_0, \\
& 0 \le u_i(t) \le 1 \quad \forall i \in \{X, Y, Z\}, t \in [0, T] \\
& \sum_{i \in \{X,Y,Z\}} u_i(t) S_i(t) \le V_0 \quad \forall t \in [0, T]
\end{aligned}
\tag{1}
$$

where $x$ refers to a state trajectory. $x(0)$ is the state vector at the initial time, i.e. at $t = 0$, and its components are non-negative. $\mathcal{S}$ refers to the collection of state trajectories conforming to the system dynamics as specified by the ODEs. The last constraint in (1) represents the vaccination capacity constraint.

The optimal control problem can be solved using standard numerical toolboxes to yield the optimal vaccination strategy. We use the numerical toolboxes Yop [41] and CasADi [42] for this purpose. Specifically, we employ the local direct collocation method to solve the optimal control by discretizing both the state and control spaces on a fine grid. The problem is then transformed into a large but sparse nonlinear program (NLP). Finally, this NLP is solved using the popular IPOPT framework (see [43] for further detail).

Finally, we rule out the need to consider "mixed" policies which split vaccine capacity between multiple groups (until at least one group runs out of susceptibles to fully utilize the vaccine capacity). We prove that *there exists an optimal vaccination strategy that devotes full capacity to a single group while the susceptibles of each type exceed the vaccine capacity. Only after the number of susceptibles decreases below capacity will such a policy ever vaccinate two groups simultaneously* (see S3 Appendix) [44]. This analytical result yields a significant reduction in the set of optimal strategies which must be considered. This reduction is also of practical importance: strategies which focus on one group at a time can be deployed more easily and align with the common practice of phased vaccine rollout to target groups.

## 3 Results

### 3.1 Model validation

We first validate our model by demonstrating its applicability to COVID-19 infection and fatality data over a period of 9 months (1 April 2020—1 January, 2021, the period preceding the introduction of vaccines). We show that despite segmenting the population into only three groups (baseline, high risk, and high contact), when the parameters are chosen correctly, our model is able to accurately predict both the spread of infection and mortality of COVID-19 as observed over the course of the pandemic in all the US states and 139 countries. We also demonstrate that any decrease in the number of groups substantially undermines the accuracy of prediction.

We consider publicly available data on infection and death counts [45]. From census and survey data, we estimate the size of the three groups. Elderly populations are categorized as high risk while specific service industry and essential workers are high contact [46–48]. See S4 Appendix for further detail on how parameters were estimated from available data. Disease parameters are obtained from WHO, CDC, and Johns Hopkins (see S1 Appendix).

We select the contact rates within each group and across groups using regression, such that the mean squared normalized error (MSNE) between the infection and death counts predicted by our model and the true values of the same is minimized. The square normalized error for infection (death, respectively) on a given day is the square of the difference between the predicted and actual infection (death, respectively) counts divided by the actual infection (death, respectively) count on that day. The mean square normalized error for infection (death, respectively) is the average of the square normalized errors for infection (death, respectively) over all days. The mean square normalized error is the average of the mean square normalized error for infection and death. The division of the square error by the magnitude of the actual infection (death, respectively) ensures that discrepancies in magnitudes of infection and death counts do not imply that the errors of the quantity with significantly larger magnitude dominates that of the other. Error can be quantified in different forms, such as mean square error, mean absolute value of error etc. We chose the mean squared error for analytical convenience

as the squared error is a smooth function, specifically it is smoother than the absolute value particularly when the square and absolute values are near zero. Thus, choosing contact rates that minimize the mean square normalized error is computationally simpler than choosing those that minimize the mean absolute value of the normalized error. The minimum mean squared normalized error will be abbreviated as MMSNE. Different contact rates were chosen for periods of two months to account for changes in both government policy (e.g. start, relaxation, and end of lockdowns) and school openings which happen at low frequency.

We now compare the infection and death counts predicted by our model with the above choice of parameters and those that were actually recorded. Although the numbers that were actually observed were used to determine the contact rates in our model, the prediction using those need not match the reality, and the MMSNE may be high. This happens when the parameter space of the minimization is not large enough to capture the complexity of the pandemic progression, e.g. if the population needed to be segmented into a larger number of groups to capture the complexity. We show that this is not the case for different locations with different population demographics and government responses throughout US and worldwide. Specifically, in Fig 3 we consider 2 US states (California, Florida) and an LMIC (Bangladesh) to demonstrate that the predicted values closely match the values actually recorded, the average

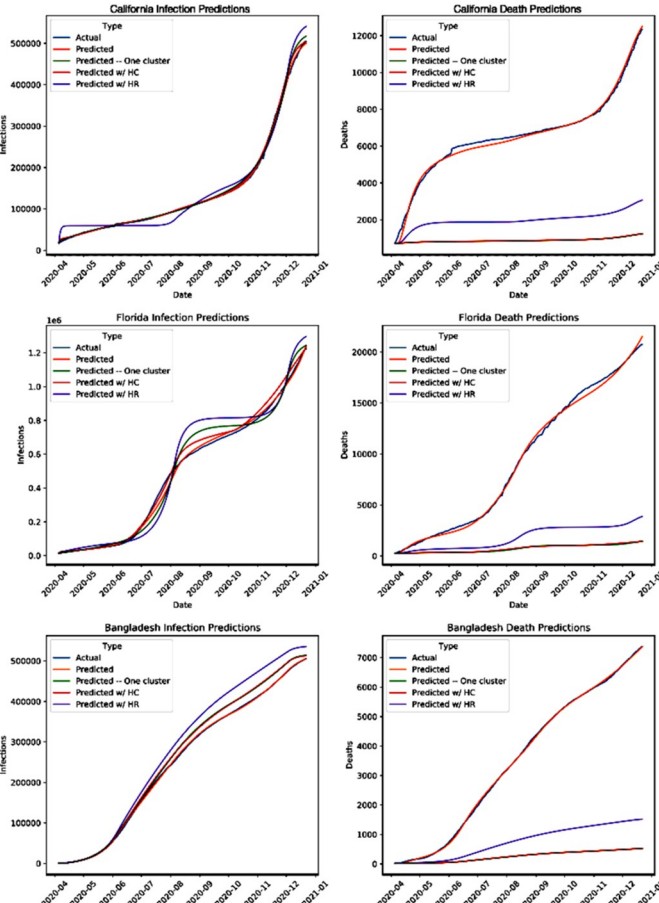

**Fig 3.** Here we show the accuracy of our model ('Predicted') when applied to the real infection and death counts of (a) California, (b) Florida, and (c) Bangladesh. The blue curves show the ground truth data [45]; orange and green show the predictions from our contact model and from a fully homogeneous model, respectively. Finally the red (purple, respectively) shows the predictions when only the heterogeneity in contact rate (risk, respectively) is accounted for.

difference between the predicted and actually recorded values is less than 1% for both infections and death. From the US, we choose California and Florida because these two states have divergent age demographics and government responses. Florida has a significantly older population as well as among the most laissez-faire NPI restrictions while the opposite is true of California [49]. We choose Bangladesh as an example of an LMIC—in general LMICs have significantly younger population demographics and different government responses as compared to the US. The MMSNE for California, Florida, and Bangladesh are very small, 0.0005, 0.001, and 0.001, respectively. Thus, our reasonably simple model has distilled the essential heterogeneity innate to the system.

We now consider all the states of US and 139 different countries surveyed. Here, four countries for which death and infection counts are available are excluded due to apparent irregularities in the available data (see S4 Appendix for further detail). As seen in Fig 4, the MMSNE was very low, below 0.006, for all US states. Across the MMSNE for all countries, the median, mean, first and third quartile are 0.002, 0.005, 0.0006, 0.005, 0.05. While this error is slightly higher than that of the states, it is still quite low. The increased error is mostly due to low population sizes and abrupt, non-smooth changes in infection and death counts (further detail in S4 Appendix). The low fitting error across the board shows that our model captures reality of geographical locations all over the world.

We now examine if our model can be simplified further while retaining the quality of the fit between the predicted values and reality. We again study California, Florida, and Bangladesh. We first consider the model that assumes a homogeneous populace—that is, there is only one group. There is therefore only one contact rate, that within the group. The mortality risk for the homogeneous populace is chosen as average over all age groups. We determine the contact rate in different time periods using the same regression technique. Fig 3 shows that the predicted counts do not closely match the counts observed in reality; the average error in predicted infections was 1.86% and the average error in predicted deaths was 82.91%. Thus, a homogeneous model can at best match only one count, and greater degrees of freedom are

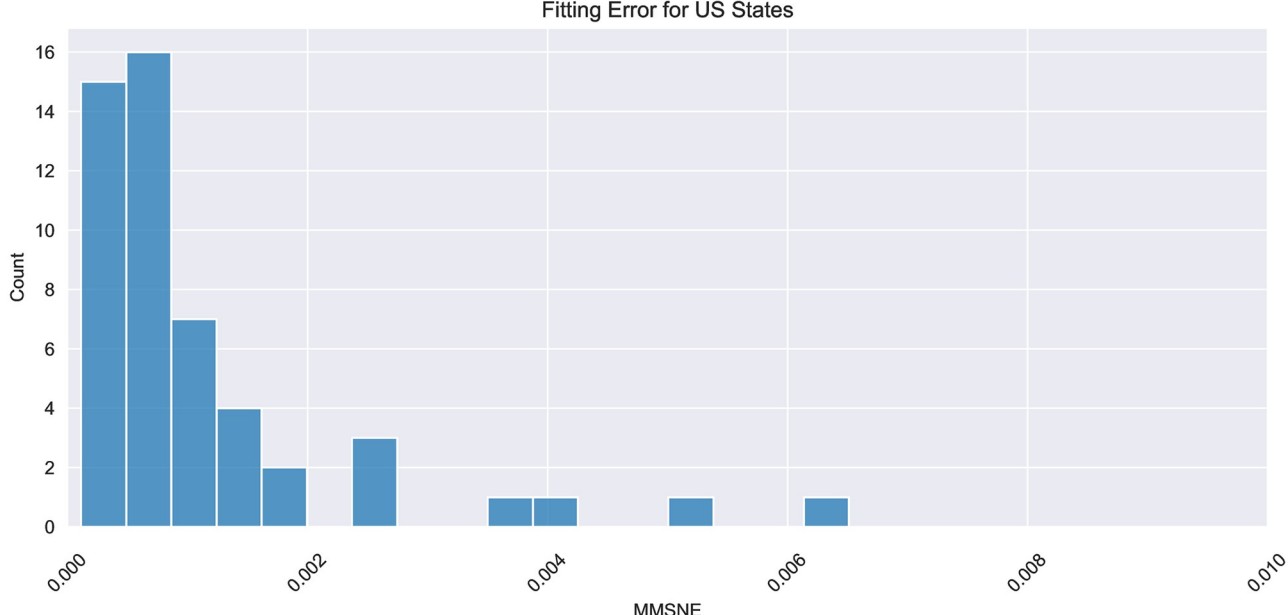

**Fig 4. This figure shows the MMSNE histogram of the predictions from our fitted model when compared to actual case and death counts ([45]) for all US States.**

necessary to match both counts. We now examine if increasing the number of groups to two provides the necessary degrees of freedom. We consider two groups with different contact rates but uniform mortality risk throughout the populace (which is average over the entire populace). The mismatch between predicted and recorded values remains similar, the average differences for the infection and death counts are respectively 1.93% and 82.85%. We now consider two groups with uniform contact rates throughout but one has higher mortality risk. Now, average error for the infection count increased to 6.95% while average death error decreased to 66.46%. Thus, all three groups are necessary for error values to be low, as three groups allows for heterogeneity in both mortality risks and contact rates. The heterogeneity innate to the evolution of the pandemic over a diverse population can not in general be captured by models simpler than ours. Having validated the model as above, we henceforth utilize the values of the parameters we extracted for three groups for all subsequent numerical computations using the model.

## 3.2 Structure of optimal vaccination policies

Unless otherwise specified, we assume that our objective is to minimize the total death count. Recall that the challenge in determining optimal policies is that the space of potential vaccination strategies is very large—it can include straightforward strategies which vaccinate groups sequentially, complex time-dynamic policies which change focus day-to-day, and policies which split vaccine capacity among multiple groups at once. But our theoretical results summarized below in Theorem 4.1 rule out the need to consider the last class of policies. Using extensive numerical computations we show that the optimal policies lie in the first class in most of the cases that arise in practice, and the suboptimality of the strategies in the first class is limited even when the optimal strategy lies elsewhere.

For a policy $u(t)$, we will use $U(t)$ to denote the vaccine allocation in absolute quantity rather than fraction i.e.

**Theorem 4.1**. *There exists an optimal vaccination strategy u such that, for each t there exist a* $(t)$, $b(t)$, $c(t) \in \{X, Y, Z\}$ *all distinct and* $U_{a(t)}(t) = min(V_0, S_{a(t)}(t))$, $U_{b(t)}(t) = min(S_{b(t)}(t), V_0 - U_{a(t)}(t))$, *and* $U_{c(t)}(t) = min(S_{c(t)}(t), V_0 - U_{a(t)}(t) - U_{b(t)}(t))$.

First, we clarify and interpret this result.

Note that for each vaccination strategy there exists a first time $t_0$ at which $S_i(t) < V_0$ for some $i$ (until $t_0$ all groups have enough susceptible individuals to be vaccinated at full capacity if they are accorded that capacity), and a first time $t_f$ at which $\sum_i S_i(t_f) \leq V_0$ (i.e. the total number of susceptible individuals is below vaccine capacity). These times are different for different strategies. Theorem 4.1 essentially says that there exists an optimal strategy $u$ which follows the following structures before its $t_0$, between its $t_0$ and its $t_f$, and after its $t_f$.

1. For each $t < t_0$, there exists some $a(t) \in \{X, Y, Z\}$ such that $U_{a(t)}(t) = V_0$, $U_j(t) = 0$, for $j \neq a(t)$. That is, at any given time $a(t)$ is the highest priority group and it is accorded the full vaccine capacity and the rest of the groups get 0 vaccine capacity.

2. For intermediate $t$ ($t_0 \leq t \leq t_f$), there exists $a(t)$, $b(t)$, $c(t)$, such that either $U_{a(t)}(t) = V_0$, $U_j(t) = 0$, for $j \neq a(t)$, or $U_{a(t)}(t) = S_{a(t)}(t)$, $U_{b(t)}(t) = V_0 - S_{a(t)}(t)$, $U_{c(t)} = 0$, or $U_{a(t)}(t) = S_{a(t)}(t)$, $U_{b(t)}(t) = S_{b(t)}(t)$, $U_{c(t)} = V_0 - S_{a(t)} - S_{b(t)}$. That is, at a time $t$, $a(t)$, $b(t)$, $c(t)$ constitute groups in decreasing order of priority. The highest priority group $a(t)$ is given as much vaccine as possible, limited only by its availability of susceptibles. The residual capacity is passed to the next highest priority group, limited only by its availability of susceptibles, and this process continues.

3. For each $t > t_f$, $U_i(t) = S_i(t)$ $\forall i \in \{X, Y, Z\}$. That is, in the final phase, the total number of susceptibles of all groups together have fallen below the vaccine capacity, and therefore all groups are allocated the maximum possible vaccine capacity they can utilize, that is, vaccine capacity equaling their susceptibles. Note that this particular property holds for all optimal policies for times after their $t_f$ (ie, even when the optimal policy is not unique).

**Remark 4.1.1**. *The salient point of this theorem is that at any given time the optimal policy described above accords different priorities to different groups; as long as the highest priority group has enough susceptibles to utilize the full vaccine capacity, it allocates the entire vaccine capacity to that group and the rest are not allocated any capacity. The policy starts splitting the capacity among different groups only when the highest priority group at the given time does not have enough susceptibles to utilize the capacity. Even then, the highest priority group is given the maximum capacity it can utilize, the residual is given to the next highest priority group limited only by its availability of susceptibles, and so on.*

**Remark 4.1.2**. *It is useful here to elaborate the extent of the above theorem. First, the theorem shows that such an optimal policy exists, but this optimal need not be unique. However, because such an optimal exists, the planner cannot achieve any further reduction in mortality by implementing more complex policies which instantaneously split vaccination capacity between two or more groups even when each group has enough susceptibles to utilize the entire vaccination capacity. Furthermore, while the specified optimal policy vaccinates one priority group at each instant (while it has enough susceptibles), our analytical result does not specify how frequently this priority group may change. We numerically study the number of switches in the priority accorded to the groups in the remainder of this section and find a maximum of 3 switches.*

To numerically study optimal vaccination policies, we vary disease parameters, initial population seroprevalence, and vaccine efficacy parameters over representative ranges based on best available estimates [45, 50–52]. Detailed derivations of all parameters and relevant sources are included in S1 and S4 Appendices. We obtain the demographic data from census as in the previous section. The default assumption is that initial infections are seeded only in the baseline group; we explicitly specify when we deviate from the default assumptions. We used the contact rates obtained from the real evolution of the pandemic as in the previous section. We also use additional contact rates from ranges of contact rates within and across age groups in 139 countries obtained from surveys and the sizes of the different groups [17]. Overall, the large range of contact rates we consider capture varying degrees of implementation and compliance with non-pharmaceutical interventions (NPIs) such as social distancing and lockdowns in different US states and the 139 countries. Over all these parameter ranges, our model was instantiated and run on a fine grid of approximately 911, 250 settings (see S4 Appendix). We also demonstrate that our model is robust to parameter estimation errors (see Section 3.6).

In 93.4% of the cases we considered, the optimal solution was either to (1) vaccinate the high contact group fully, next vaccinate the high risk group fully (2) vaccinate the high risk group fully, next vaccinate the high contact group fully. In both these cases, the baseline group is vaccinated at the end. We refer to the first as *high contact prioritization* and the second as *high risk prioritization*. In the rest of the cases there is an extra step where the optimal policy switches from high contact vaccination to high risk before returning to the high contact. In these few cases, the better performer of the above two simple policies had only 2.2% more deaths than the optimal policy. Even including these cases, across all numerical experiments, the priority group for vaccination changed at most three times over the course of the optimal policy, and the number of switches was three in only 6.4%. Coupled with our theoretical result, this simplifies our potential policy space down to two easily deployable policies even without any assumed restrictions on policy complexity.

Seeing that the aforementioned simple policies perform well, one may question whether all simple policies, even static ones, perform similarly well in reducing mortality. To answer this question, we compare the death counts of the overall optimal policy with two more restricted policies: (1) the optimal policy among those whose vaccine capacity allocations to the groups are constant over the entire time horizon (*static policy*) and (2) the optimal policy among those whose vaccine capacity allocations to the groups are constant in periods of one month (*monthly policy*). The first policy is completely static while the vaccination capacities allocated by the second can vary with time but over large time scale. In contrast the overall optimal policy has been obtained by optimizing among policies that can vary their vaccination capacity allocation to groups as frequently as deemed necessary. We find that even at their best, the less dynamic classes of policies under consideration perform significantly worse than the general optimal. In particular, Fig 5a and 5b, show the additional mortality incurred by each of these restricted policies when compared to the optimal policy mortality as a percentage of the latter. The axes are measures of the two types of heterogeneity: the ratio of risk parameters of the high risk and baseline groups (x-axis) and the ratio of daily average contact of the high contact and baseline groups (y-axis). We see that as heterogeneity increases, the difference in mortality between the optimal and the two static variations is magnified. Furthermore, the suboptimality of the more static policies is especially sensitive to the high contact group. Intuitively, a very active high contact group significantly accelerates the spread of the epidemic, thus necessitating a more dynamic vaccine allocation strategy. Over all instances, the average increase in death counts under the two restricted policies as compared to the overall optimal vaccination strategy were 206.3% and 72.16%, respectively. In the extreme, the mortality of the static policy is over thrice that of the optimal. We note that the monthly policy has both lower death count and is more dynamic than the static policy. The most extreme difference between the two restricted policies occurs when the high contact multiple is high i.e. when the high contact group is very active. In this regime the monthly policy far outperforms the static policy (while both remain significantly less effective than the true optimal). Intuitively, a very active high contact group significantly accelerates the spread of the epidemic, thus necessitating a more dynamic vaccine allocation strategy.

Motivated by the above observations, we investigate the importance of policy dynamism for mortality reduction (Fig 5c). We consider one specific instance for this purpose: the US baseline setting in which contact rates and population demographics are averaged over the data reported in different states [17, 47, 53]. Fig 5c depicts how mortality substantially decreases as we increases the number of decision intervals, expanding our optimization space to allow for more dynamic policies. The dynamism of our optimal policies is integral to their performance in reducing mortality. If such dynamic policies are ignored, we do not have a proper foundation for reasoning about the efficacy of vaccination policies because such restrictions preclude identifying the minimum possible death count.

A strength of our framework is its computational tractability. The mean computation time needed to determine the optimal vaccination strategy over all aforementioned parameter settings was just 10.86 seconds with a standard deviation of 39.88 seconds. In less than 0.6% of instances, the computation time exceeded the maximum allotted time of 500 seconds. Excluding these outliers, the mean computation time was just 8.04 seconds with a standard deviation of 14.45 seconds and a maximum of 142.42 seconds (a histogram of runtimes can be found in the S5 Appendix). These computation times were obtained using modest computational resources, namely an Intel i7 2.7 GHz machine with 16 GB RAM, which could also be readily available in local public health bodies and resource-constrained settings such as LMICs.

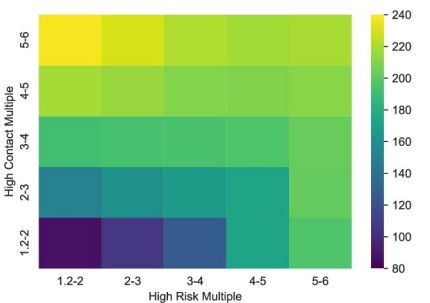

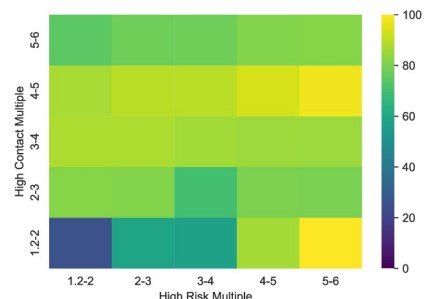

**(a)** The increase in mortality caused by static vaccination policies over the optimal vaccination policy

**(b)** The increase in mortality caused by policies that change allocations only monthly over the optimal vaccination policy

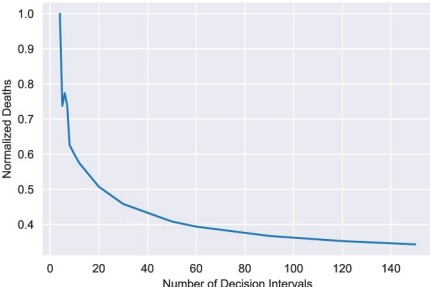

**(c)** The benefit of more dynamic vaccination policies.

**Fig 5.** Fig 5a shows the increase in the death counts of the static vaccination policy over that of the optimal policy as a percentage of the latter. The x-axis is the ratio of death rates of the high risk and baseline groups (this is also the ratio of the symptomatic rates and hospitalization rates of the two groups). The y-axis is the ratio of daily average contacts of the high contact and baseline groups. As these ratios increase, our contact network becomes more heterogeneous and the suboptimality of the static policy increases. Fig 5b shows the same for the monthly policy, i.e. when the vaccine rate allocation to groups is allowed to change only on a monthly basis. Fig 5c shows the decrease in the number of deaths in the US baseline instance as one increases the number of decision intervals. The allocation of the vaccination capacity to groups remain constant over each interval. Number of days in each interval equals 365 divided by the number of intervals. Thus, more decision intervals allow for more dynamic policies. The death count is normalized by the maximum number of deaths across all data points. (**a**) The increase in mortality caused by static vaccination policies over the optimal vaccination policy (**b**) The increase in mortality caused by policies that change allocations only monthly over the optimal vaccination policy (**c**) The benefit of more dynamic vaccination policies.

## 3.3 Comparison of mortalities under high contact and high risk prioritization across different parameter ranges

It is noteworthy that most public health authorities implemented a strategy closest to high risk prioritization with the caveat of early vaccination for health care workers. High contact prioritization, on the other hand, was not often recommended, especially with the broader definition of high contact which we use. Yet, in the previous section we found that the high contact prioritization was optimal in several realistic settings. In this section we compare the mortalities under high contact and high risk prioritization policies by varying different combinations of parameters. We find that the high contact prioritization has substantially lower mortality than the high risk prioritization in 42% of instances; thus the prevalent norm has been suboptimal in many cases.

To compare the two strategies, in each case, we vary two parameters as shown in the two axes of Figs 6 and 7, for given choice of one variable and include the relative decrease of

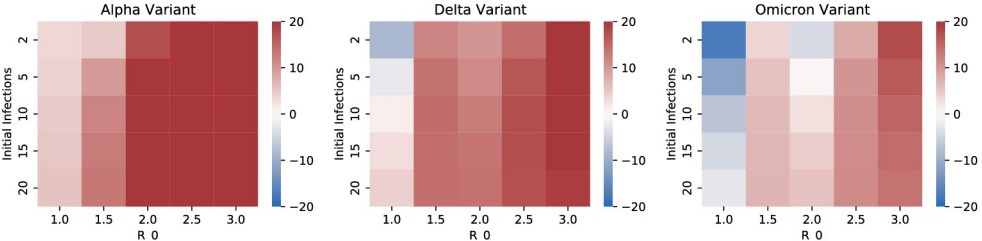

**(a)** The effect of different COVID-19 variants on optimal vaccination policy

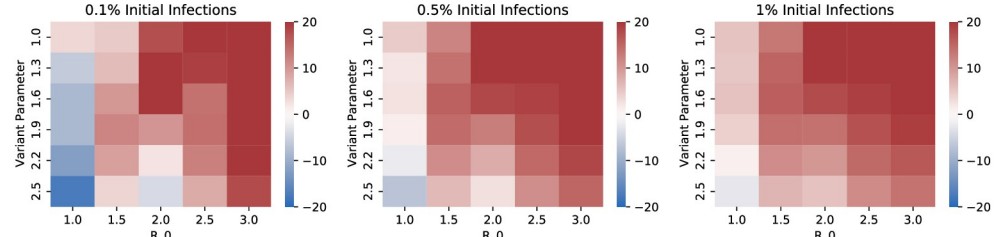

**(b)** The effect of initial seroprevalence on optimal vaccination policy



**(c)** The effect of $R_0$ on optimal vaccination policy

**Fig 6. Each heatmap displays the difference in resultant mortality between high contact prioritization vaccination and high risk prioritization as a percentage of the mortality of high risk prioritization policy.** Fig 6a–6c, respectively, demonstrate the impact of 1) variants with different transmissibility and mortality characteristics 2) different values of initial seroprevalence 3) different values of $R_0$. (**a**) The effect of different COVID-19 variants on optimal vaccination policy (**b**) The effect of initial seroprevalence on optimal vaccination policy (**c**) The effect of $R_0$ on optimal vaccination policy.

mortality of one policy over another for all values of other parameters as considered in Section 3.2. We evaluate the death count of any given policy by utilizing the system of ODEs in Fig 2 with vaccination rates $\{u_i(\cdot)\}$ that correspond to the policy. The default assumption is that the variant is Alpha; we explicitly specify when we deviate from the default assumptions. After all other parameters are set, $R_0$ is computed via the next generation matrix method (see [54]) and contact matrices are normalized to achieve the desired $R_0$, whenever we choose values of $R_0$. Refer to S4 Appendix for further details.

Fig 6 shows the effect of COVID variants, level of initial infection and $R_0$ on the mortalities under the two policies. We considered the three major variants thus far (Alpha, Delta, and Omicron) by choosing the transmissibility (probability of transmission upon contact with an infective individual) and mortality characteristics according to recent clinical data for each strain [55]. In particular, relative to the Alpha variant, Delta was nearly twice as contagious and with double the hospitalization risk while Omicron was more contagious still but with

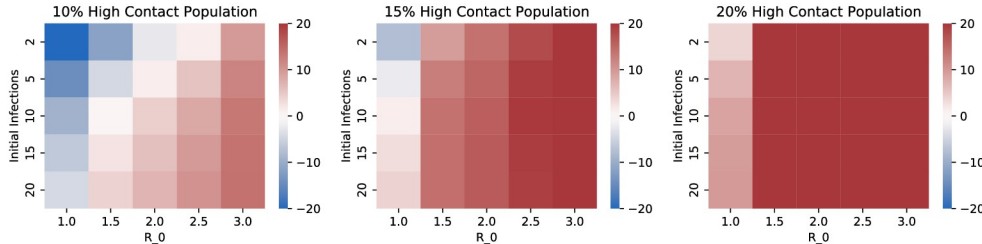

**(a)** The effect of demography on optimal vaccination policy

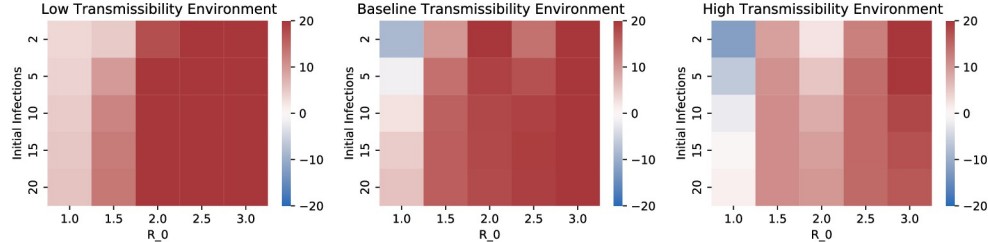

**(b)** The effect of transmissibility on optimal vaccination policy

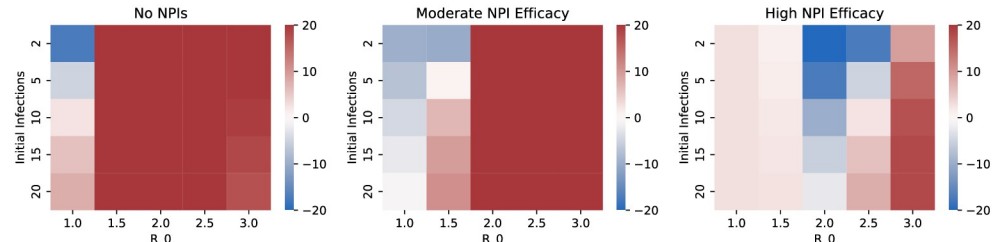

**(c)** The effect of NPI Efficacy on optimal vaccination policy

**Fig 7. Each heatmap displays the difference in resultant mortality between high contact prioritization vaccination and high risk prioritization as a percentage of the mortality of high risk prioritization policy.** Fig 7a–7c, respectively demonstrate the impact of 1) different population distributions, i.e. different values of the percentage of the total population that is in the high contact cluster 2) transmissibility of the virus 3) NPIs (e.g. social distancing, masking, lockdowns). (**a**) The effect of demography on optimal vaccination policy (**b**) The effect of transmissibility on optimal vaccination policy (**c**) The effect of NPI Efficacy on optimal vaccination policy.

lower mortality rates [56]. The variant parameter in Fig 6b and 6c is applied as a multiplicative factor on transmissibility—this range allows us to consider potential future variants of concern. Fig 6 shows that, among the three variants, Omicron has the highest relative benefit of high contact prioritization and Alpha the least. This happens because Omicron and Alpha respectively have the highest and least transmissibilities. And, vaccinating high contact individuals reduces the overall transmission significantly which is particularly effective in decreasing the overall mortality when the virus has higher transmissibility. We also note that the relative benefit of high contact prioritization decreases with increase in the level of initial infection. This happens because if the initial infection is high the virus spreads in the high risk group during the initial time in which the high contact group is being vaccinated, which in turn substantially increases the mortality of the high risk group because of the innate high mortality rates of the infected in this group. The effect of $R_0$, however, was non-monotonic with high risk prioritization attaining lower mortality than the high contact prioritization at extreme values of $R_0$. We will explain the non-monotonic behavior in the next paragraph.

Fig 7 shows the impact of the high contact population size, viral transmissibility, and NPI (non-pharmaceutical intervention, i.e. social distancing, masking, lockdowns) efficacy on the two policies under consideration. Note that while the variants considered in the previous paragraph have different transmissibilities and mortality risks, here we study the impact of both these factors in isolation. We choose multiplicative factors of 1.0, 1.5, and 2.0 respectively on the baseline transmissibility to obtain the low and high transmissibility environments. NPI efficacy is modeled as a multiplicative factor on contact rates to capture different extents of social distancing, masking, and lockdowns. The low, moderate and high values of NPI efficacy respectively correspond to multiplicative factors of 1.0, 0.7, and 0.4, that is, no reduction in contacts, 30% reduction in contacts, and 60% reduction in contacts. Higher viral transmissibility increases the relative benefit of high contact prioritization due to larger reductions in potential infections. However, the relative benefit of the early vaccination of the high contact group is decreasing in the group's size. This happens because larger groups need more time to be vaccinated. Thus, according vaccine priority to large sized high contact groups significantly delay vaccination to high risk groups; the disease may spread in the high risk group during this delay leading to high death counts owing to higher mortality rates therein. Note that $R_0$ is increasing in both transmissibility and size of high contact groups. Thus, an increase in $R_0$ can shift the relative benefits of high contact prioritization policy in either direction.

Finally, the range in Figs 6 and 7 which exceed 20% difference or fall below -20% difference between the two policies includes values considerably higher than 20% or lower than -20%. Fig 8 shows the histogram of percent differences in mortality between the two strategies. Specifically, for certain parameter values in the above range, the percentage difference was as high as 71.71% and as low as -81.85%. Overall, the numerical computations reveal that, while the common high risk priority policy is optimal in the majority, there is a considerable range of realistic parameters for which high contact prioritization more effectively lowers mortality.

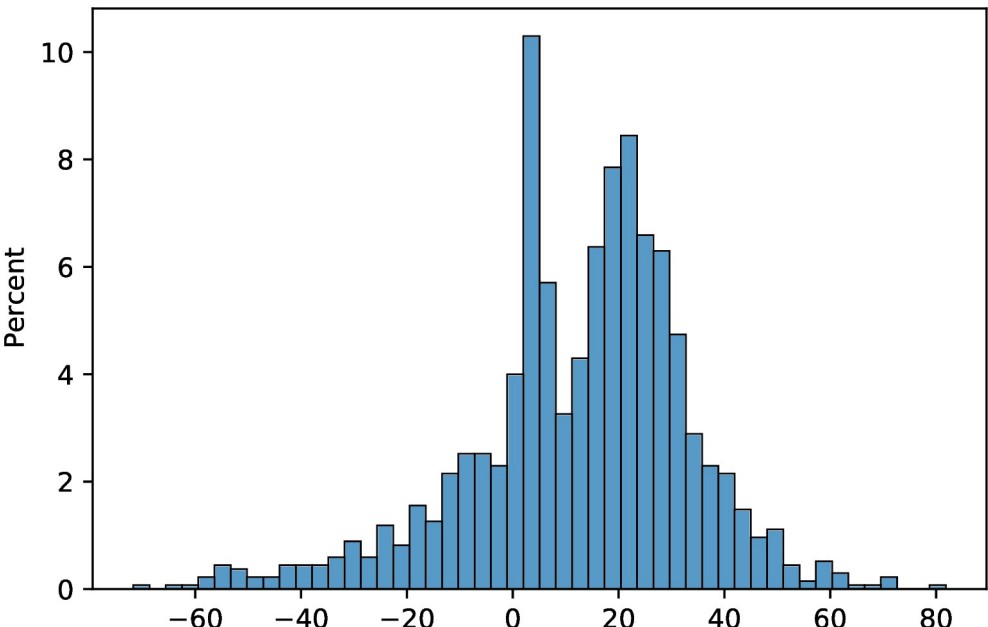

**Fig 8. Histogram of the percent difference in mortality between high contact prioritization and high risk prioritization vaccination strategies over all instances.**

### 3.4 Case studies

We now compare the death counts of the high contact and high risk prioritization vaccination strategies and a benchmark vaccination strategy for three important case studies which arise in practice. These cases: (1) LMICs, (2) prisons, (3) long-term care facilities, have different characteristics, suffer from high vulnerability to the disease, and are hotbeds for the generation of new variants [57–59]. In particular, LMICs have a smaller elderly population, but, due to the greater prevalence of inter-generational households, high risk individuals are less able to isolate. They also have generally lower vaccine capacity [3]. Prisons also have a smaller elderly population, but have high levels of transmissibility due to unhealthy living conditions and presence of a large number of individuals in a relatively small confined space. Long-term care facilities have a large elderly population alongside a small core group of employees that have high contact rates with those at high risk. These differences were modeled by setting appropriate vaccination capacity constraints, contact rates between groups and sizes of the different clusters according to vaccine records, census data, government statistics, and contact surveys (for further detail see S1 and S4 Appendices) [3, 45, 60, 61]. We also consider the US baseline as described in Section 3.2. The benchmark vaccination strategy, referred to as Uniform, distributes vaccines among groups proportionally to each group's size. Uniform is similar to vaccinating randomly chosen individuals subject to vaccination capacity constraints.

Fig 9a and 9b shows the differing efficacy of vaccine policies between the US baseline model and LMIC. Unlike the US baseline, the high contact prioritization strategy minimized mortality in the LMIC setting among the policies considered. This is because high risk individuals are in more frequent contact with the high contact group in LMICs than in the US. Thus, the high contact group constitutes a greater vector of spread to the high risk group in LMICs than in the US, and vaccinating the former on a priority basis helps protect the latter. Fig 9(a) and 9(b) also show how vaccine scarcity enhances the impact of right vaccine prioritization. Note that 0.5%, 0.2% of overall populace can be vaccinated every day in US baseline and LMIC respectively. We see that the difference in infections, hospitalizations, and YLLs between the best performing strategy and the alternatives is much greater in the LMIC than in the US Baseline. Even when considering deaths, the difference between uniform and high contact prioritization is much greater in the LMIC.

Next, due to the large high risk population, long-term care facilities have relatively high mortality rates [62]. Even with high individual mortality, high contact prioritization more effectively minimizes mortality (Fig 9(d)). Intuitively, because the high risk group is large in this setting, vaccinating the group fully would be a lengthy process during which the infection can spread among the group. Thus, cutting off the high contact group as a vector of the spread more effectively reduces mortality. The opposite is true in the prison setting because the high contact group is large and viral transmissibility is high due to living conditions [63]. Both of these ensure widespread infections. The best recourse is to vaccinate the small size high risk group first, this can be accomplished in a short time (Fig 9 (c)).

In all settings, the better of the high risk and high contact prioritizations, reduces mortality more than uniform vaccination. But the worse of the two prioritizations is sometimes worse than uniform vaccination. This is the case in the US baseline and nursing home as seen in Fig 9(a) and 9(d). Simply using one of the two prioritizations in all cases, without verifying which one has lower mortality count, incurs higher mortality even compared to uniform (i.e. random) vaccination in some cases. Thus, one must tailor the prioritization to the specific context.

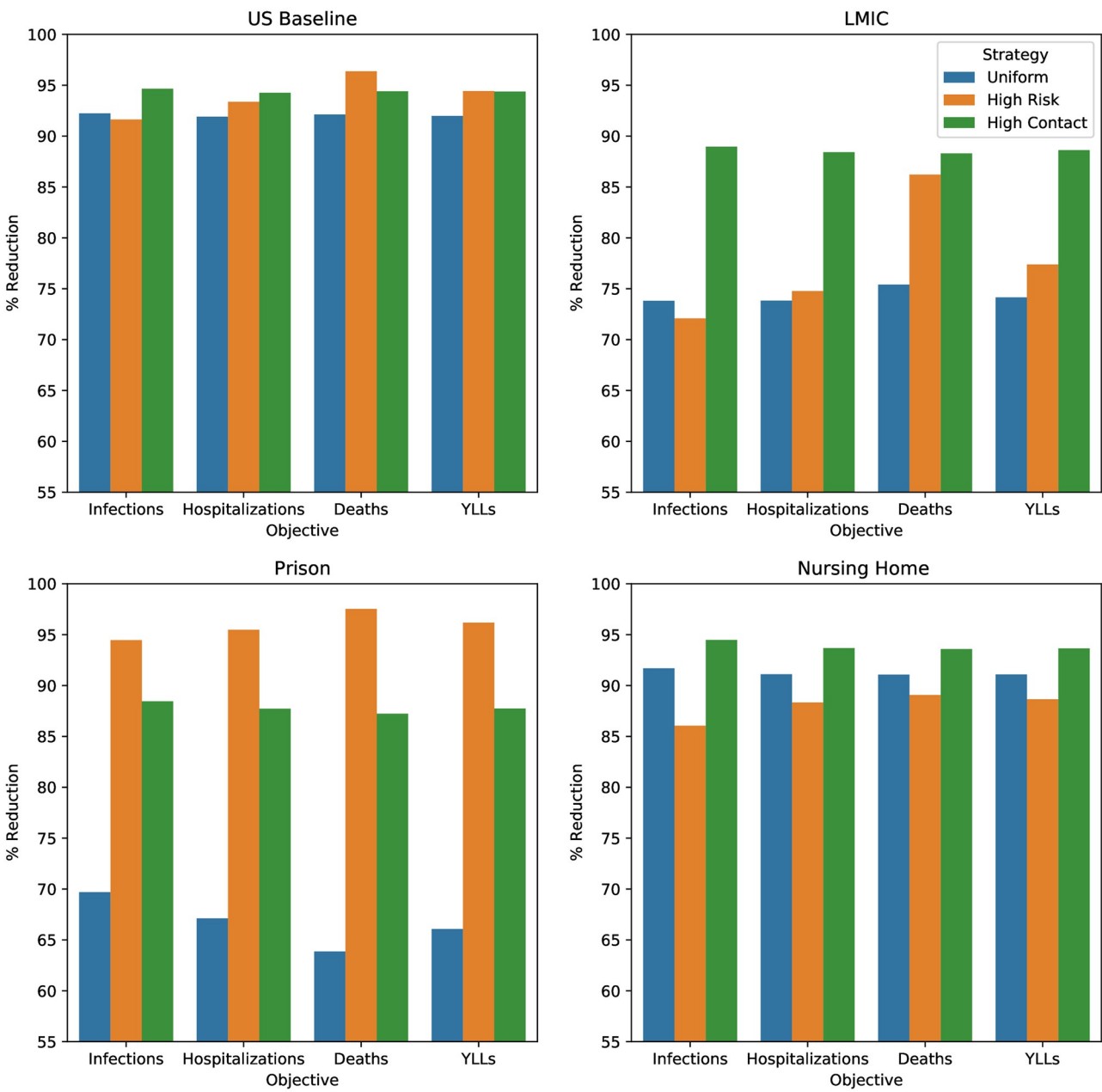

**Fig 9. Each figure compares the infections, hospitalizations, deaths, and years of life lost (YLLs) under uniform (blue), high risk (orange), and high contact (green) prioritization vaccination policies.** We depict the percentage reduction of these attributes under each policy relative to the no vaccination scenario. Fig 9a (top left) is instantiated on US demography, NPI efficacy, and topology. Fig 9b (top right) is instantiated on an LMIC model with lower elderly population, greater high contact population, and higher transmissibility. Fig 9c (bottom left) is instantiated on a prison model comprised of guards (high contact), facility employees (baseline), and prisoners (baseline, high risk). This environment corresponds to low elderly population and high transmissibility. Fig 9d (bottom right) is instantiated on a nursing home model with a large elderly population (high risk), medical staff (high contact), and administrative staff (baseline).

## 3.5 Sensitivity analysis

We now analyze how changes in specific model parameters affect the optimum (ie, minimum) total death count and optimal policy. We consider three modeling attributes: contact rates, group sizes, and vaccine efficacy. We fix the baseline contact matrix to be that of the US and

vary a multiplicative factor applied on the contact rates in the baseline contact matrix (referred to as contact rate multiplier). We also vary the COVID variant, initial seroprevalence, and high contact population sizes as previously outlined in Sections 3.2 and 3.3 (see S4 Appendix for further detail).

To study the impact of varying the contact rate multiplier, we instantiate our model on contact rate multipliers in the range [0.4, 1] at intervals of 0.075. The range was chosen to cover the estimates of contact rate reduction based on COVID countermeasures such as masking, social distancing, and lockdowns [64]. For each value of the multiplier, we vary the COVID variant, initial seroprevalence, and high contact population sizes and record 1) the percentage of instances in which the high contact prioritization policy is optimal 2) the average of the optimum value of the objective function (ie total death count)(Fig 10a and 10b). We similarly study the impact of the high contact group size by varying the values of the COVID variant, initial seroprevalence, and contact multiplier for a given group size and record 1) and 2) (Fig 10c and 10d). We vary the size of the high contact group size while keeping the size of the high risk group and size of the overall population constant, thus an increase in the size of the high

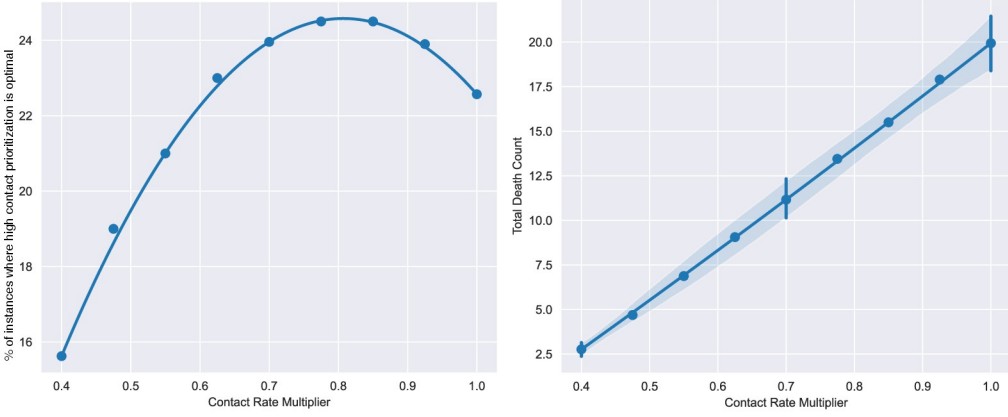

**(a)** The effect of contact rates on the optimal policy

**(b)** The effect of contact rates on expected minimum total death count per 1000

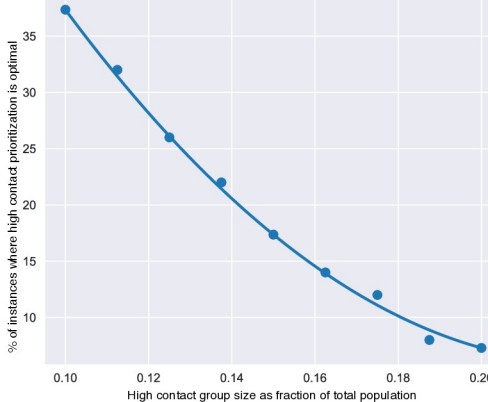

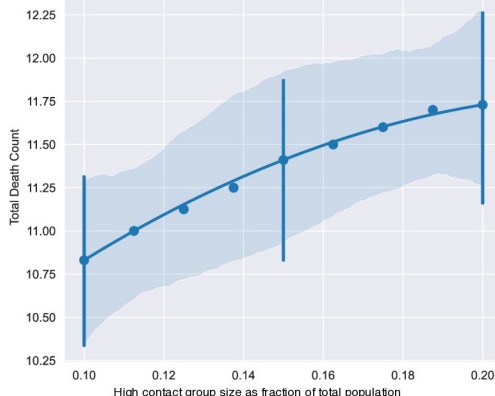

**(c)** The effect of group size on the optimal policy

**(d)** The effect of group size on the expected minimum total death count per 1000

**Fig 10. Error bars indicate an interval of a 50% confidence interval.** (**a**) The effect of contact rates on the optimal policy (**b**) The effect of contact rates on minimum total death count per 1000 (**c**) The effect of group size on the optimal policy (**d**) The effect of group size on the expected minimum total death count per 1000.

contact group size implies a commensurate decrease in the size of the baseline group. High contact group size is varied in the range [10, 20]% of the total population. This range was chosen to cover the values of the states and countries for which we have data—for reference, the US high contact population is estimated to be approximately 16% [65].

As expected, Fig 10b shows that the total death count is increasing as the multiplier increases. However, the prevalence of the high contact prioritization as the optimal policy shows non-monotonicity with respect to the contact rate multiplier (see Fig 10a). Extreme values of the multiplier lead to improved performance of high risk prioritization. Intuitively, low contact rates indicate that the disease spreads slowly. Thus, there is little utility in attempting to decrease the spread further. When contact rates are high, the disease spreads very quickly. Thus, it will become widespread before it is possible to cutoff the spread by vaccinating the high contact group. Thus, high contact prioritization reduces mortality the most for intermediate values of the multiplier. This non-monotonicity mirrors the results depicted in Fig 6c which have been discussed in Section 3.3. We note that the mean optimal total death count shows considerable sensitivity to contact rates: it varies by 700%, from approximately 2.5 to 20 deaths per thousand.

Fig 10c shows that the number of instances in which high contact prioritization is optimal decreases in the group's size. This happens because larger groups need more time to be vaccinated. Thus, according vaccine priority to large sized high contact groups significantly delay vaccination to high risk groups; the disease may spread in the high risk group during this delay leading to high death counts owing to higher mortality rates therein. In Fig 10d, we see that the mean death count changes only by 8.3%, from 10.8 to 11.7 deaths per thousand. The death count therefore shows relatively modest sensitivity to the size of the high contact group. This happens because as the high contact population increases, the optimal policy adjusts accordingly and is able to avoid significant changes in the death count.

We also investigate the sensitivity of the death count and optimal policy to vaccine efficacy. Recall that vaccination has two primary goals: (1) reducing the severity of infection for vaccinated individuals upon contracting the disease (i.e. *protecting*) and (2) reducing the vaccinated individuals' probability of contracting the disease (i.e. *preventing*). If a vaccine reduces only the probability of hospitalization and/or death from late infection stage, then it only protects. If it reduces only the recipient susceptible's probability of contracting the disease upon being in contact with an infectious individual, it only prevents. If it reduces the probability that a recipient develops symptomatic infection upon contracting the infection, it protects and also indirectly prevents. This is because an asymptomatic individual does not develop symptoms (by definition) and naturally is neither hospitalized nor dies. The probability of transmission from the asymptomatic state is lower than from the symptomatic state. Thus, if a vaccine increases the proportion of asymptomatic infection, it reduces further transmission (and hence we use the characterization that it indirectly prevents). Usually vaccines both protect and prevent but often with different efficacy. It is noteworthy that current COVID vaccines drastically decrease the probability of symptomatic infection, hospitalization, and death; and, to a lesser degree, decrease the probability of contracting the disease [56].

Guided by the above observations, we parameterize vaccine efficacy along the four possible modes of action—reduction in probability of a vaccinated susceptible contracting the disease upon coming in contact with an infectious individual, probability that a vaccinated individual develops symptomatic infection upon contracting the disease, probabilities of hospitalization, death conditioned on developing symptoms. Let an unvaccinated individual's *risk profile* $r_0 = \begin{bmatrix} a & b & c & d \end{bmatrix}$ respectively be the above four probabilities. For a given vaccine, we define its *efficacy profile* $v = \begin{bmatrix} w & x & y & z \end{bmatrix}$ such that a vaccinated individual will have risk profile

$r_v = \begin{bmatrix} a(1-w) & b(1-x) & c(1-y) & d(1-z) \end{bmatrix}$. Thus, a vaccine with efficacy profile $\begin{bmatrix} 1 & 1 & 1 & 1 \end{bmatrix}$ provides perfect protection and prevention.

We consider four different models of vaccine efficacy characterized by an efficacy parameter $\pi$: (1) a multiplicative decrease in probabilities of contraction, symptomatic infection, hospitalization, and death ($v = \begin{bmatrix} \pi & \pi & \pi & \pi \end{bmatrix}$), (2) a multiplicative decrease in only probability of contraction ($v = \begin{bmatrix} \pi & 0 & 0 & 0 \end{bmatrix}$), (3) a multiplicative decrease in the probabilities of symptomatic infection, hospitalization, and death ($v = \begin{bmatrix} 0 & \pi & \pi & \pi \end{bmatrix}$), and (4) a multiplicative decrease in the probabilities of hospitalization and death ($v = \begin{bmatrix} 0 & 0 & \pi & \pi \end{bmatrix}$). Model (1) protects and prevents, model (2) only prevents, model (3) protects and indirectly prevents, model (4) only protects. Since current COVID vaccines drastically decrease the probability of symptomatic infection, hospitalization, and death; and, to a lesser degree, decreases the probability of contraction [56], it is perhaps closest to protection and indirect prevention.

For each model of vaccine action, we instantiate parameter $\pi$ in the range [0.7, 0.9], covering the ranges indicated from COVID vaccine studies [51, 52]. We fix the contact matrix to be that of the US and vary the COVID variant, initial seroprevalence, high contact population size, and contact rate multiplier as described earlier in this section. For each model and at each value of $\pi$, we record 1) the percentage of instances in which the high contact prioritization policy is optimal and 2) the average of the optimum value of the objective function (i.e. total death count) over the parameter values varied here.

Fig 11b shows that the optimal death count is the lowest for protection and prevention; this is expected as it decreases all four relevant probabilities (contraction, symptomatic infection, hospitalization, and death). For similar reasons, protection and indirect prevention (decreasing symptomatic, hospitalization, and death probabilities) has lower death count than protection only (decreasing hospitalization and death probabilities) and prevention only (decreasing only contraction probability). When comparing the death counts of prevention only and protection only, we see two regimes: when efficacy is low, the prevention only vaccine provides greater reduction in death count, when efficacy is high, the opposite is true. We also observe

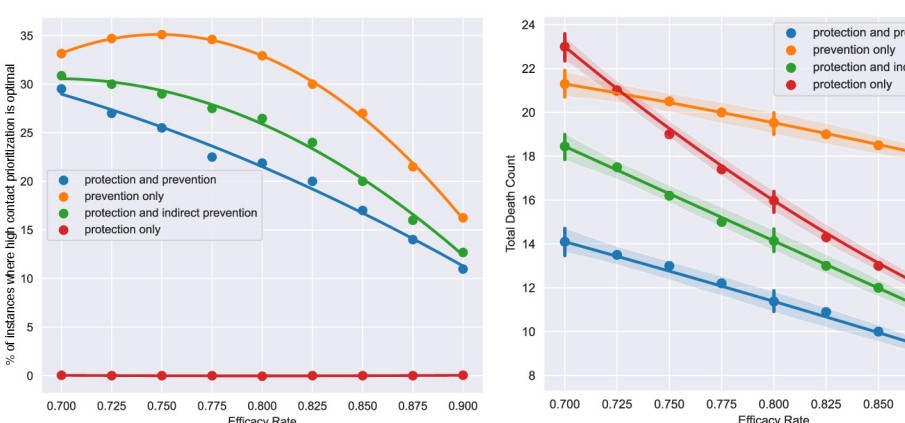

**(a)** The effect of various vaccine efficacy models on optimal vaccination policy

**(b)** The effect of various vaccine efficacy models on expected overall mortality

**Fig 11. These figures show how optimal vaccination policy (left) and mortality (right) are affected by the 4 models of vaccine efficacy (protection and prevention, protection and indirect prevention, protection only, and prevention only), respectively by column.** The top row shows the percentage of instances for which high contact prioritization was optimal. The bottom row shows the average total death count when applying the optimal policy with error bars indicating an interval of one tenth of the standard deviation. (**a**) The effect of various vaccine efficacy models on optimal vaccination policy (**b**) The effect of various vaccine efficacy models on expected overall mortality.

that the mean optimal death counts in the protection and prevention, protection and indirect prevention, protection only, prevention only models vary in intervals [8, 14.2], [9.4, 18.4], [10.9, 23.1], and [17.5, 21.3] deaths per thousand, respectively as the vaccine efficacy parameter is varied in the shown range. Thus, the protection only vaccine is particularly sensitive to efficacy rate, exhibiting the steepest decrease (52.81%) in death count in Fig 11b with increase in efficacy, while the prevention only vaccine is the least sensitive (17.84% decrease). The protection and prevention, protection and indirect prevention vaccines are also considerably sensitive to vaccine efficacy (43.66%, 48.91% decrease respectively).

In Fig 11a, we see that, for a protection only vaccine, the optimal vaccination policy never prioritizes the high contact group. This is because the protection only vaccine does not reduce the probability of contraction of the disease of the recipient and therefore does not reduce the spread. So, it seeks to reduce death by preferentially vaccinating, and therefore protecting, those most likely to die, ie the high risk ones. For vaccines that both protect and prevent (either directly or indirectly), we see that the prevalence of high contact prioritization as the optimal strategy monotonically decreases in efficacy rate; and the prevalence is higher for the protection and indirect prevention vaccine between the two. For the prevention only vaccine, we see that the number of instances in which high contact prioritization is optimal changes non monotonically with change in the efficacy parameter. This mirrors the non-monotonicity with respect to contact rates which we observe in Fig 10a.

### 3.6 Robustness study

The estimates of parameters will inevitably have some errors. We therefore investigate the robustness of the computation framework for the optimal vaccination strategy to estimation errors as mentioned in Section 3.2.

We first investigate robustness with respect to disease parameters. We inject white Gaussian noise into transmissibility and fractions of individuals who become symptomatic (from the exposed state), are hospitalized (from late stage infection) and die (after hospitalization). That is, for each such parameter, the value was multiplied by $(1 + \mathcal{N}(0, \sigma^2))$ where $\mathcal{N}(\mu, \sigma^2)$ is the normal distribution with mean $\mu$ and standard deviation $\sigma$. We ignore the few cases in which the erroneous estimates become negative. Whenever the erroneous estimate of the fractions exceeds 1, we consider the fraction to be 1. The above parameters affect transition rates into some states. We take $\sigma \in \{0.05, 0.1, 0.15\}$. Note that this noise model is a common one which has been observed in complex dynamical systems like our epidemic model [66].

While the true dynamics of the evolution were determined by the original parameter, the computation framework only had access to these noisy estimates. The optimal vaccination strategy computed using these noisy estimates is referred to as 'noisy optimal'. The average increase in death count due to utilization of the noisy optimal strategy was below 2%, although, in extreme cases it was as high as 84.6%. Between high contact and high risk policies, the policy that has lower death counts was identified correctly under the noisy parameters in 89.1%, 88.2%, and 86.2% of cases, respectively for $\sigma \in \{0.05, 0.1, 0.15\}$. Thus, overall the framework is robust to estimation error.

When evaluating the vaccination policy made with noisy information, 88.2% of all instances exhibited a suboptimality of less than 1% when the controller only had access to noisy disease parameters (noise levels of 5%, 10%, and 15%). The histogram of the suboptimality in the remaining 11.8% of cases is included below in Fig 12.

We now consider robustness with respect to errors in estimating: 1) contact rates, 2) size of the high contact group, and 3) vaccine efficacy. For robustness to errors in estimating contact rates, we apply a multiplier on the contact rate matrix as in Section 3.5 and consider errors in

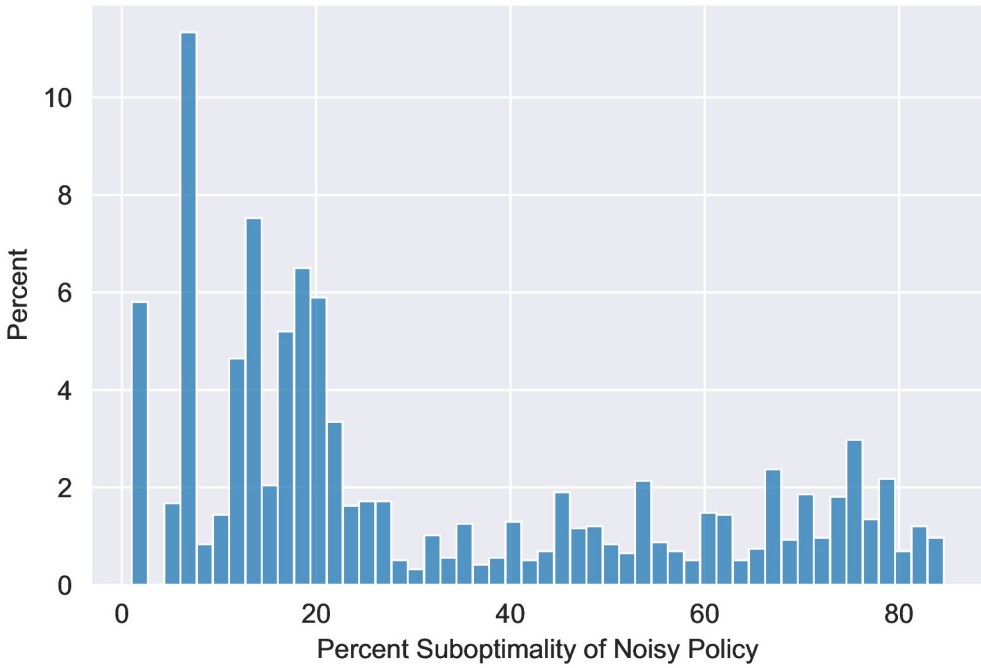

**Fig 12. Histogram of suboptimality of noisy vaccination policy when suboptimality exceeds 1%.**

estimating it. We inject multiplicative Gaussian noise into 1) the contact rate multiplier, 2) the size of the high contact group, and 3) the vaccine efficacy as we did for the disease parameters, one at a time. To be consistent with our sensitivity analysis in Section 3.5, we consider the protection and prevention vaccine model as described there, and an efficacy value of 0.75, which is towards the lower end of the realistic range. We would describe our findings for higher vaccine efficacy as well. Fig 13 shows the resulting suboptimality as a function of $\sigma$. The resulting suboptimality is small in each case. Thus the death count by and large only increases marginally when these parameters are estimated erroneously, although the optimum value of the death count is considerably sensitive to values of different parameters as seen in Section 3.5. We next explain why this is the case.

Robustness to error in estimation of parameters ultimately depends upon how much the optimal policy varies as a function of the parameter. Even if the optimal death count varies rapidly with small changes in a parameter, the optimal policy may be stable. For example, in Fig 11, for the protection only vaccine, as efficacy rate increases, the death count decreases rapidly (see Fig 11b), however, there is no change in the optimal policy (see Fig 11a). Thus, the policy that is optimum at the true value of the efficacy rate is computed as optimum even for the estimated value, which results in the minimum possible death count, regardless of the estimation error. Thus, such a system would be perfectly robust to estimation errors for the efficacy rate, although the optimal value of the death count is itself quite sensitive to the value of the efficacy rate.

We now investigate how much the optimal policy varies as a function of the three parameters we are now considering, to understand why the system turned out to be robust to estimation errors. We vary each of the above three parameters, one at a time, over realistic ranges as described in Section 3.5. When we vary one parameter, we fix the other two at their default values. Throughout all of the specified instances, the optimal policy was the one with lower death count between HRP and high contact prioritization (HCP). Fig 14 provides the death counts

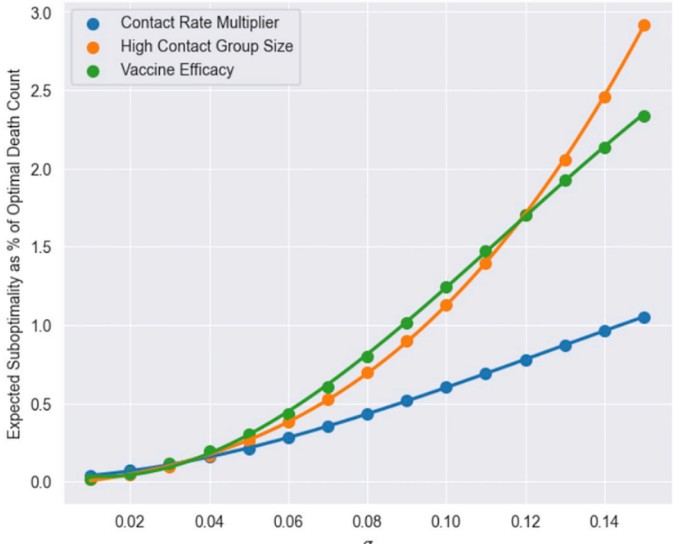

**Fig 13. We show the expected suboptimality as a function of the standard deviation $\sigma$ of our multiplicative Gaussian noise.**

under HRP and HCP, thus, the optimal vaccination policy can be easily discerned between the two at each instance. We note that the optimal policy remains the same in large segments, and switches between HRP and HCP only once in the studied range. If the true and estimated values of a parameter are in the same segment, e.g. the green star and blue circle on the x-axis of Fig 14b, they both yield the same optimum policy; in this case, even if the estimation error is large, the optimal policy is identified correctly and no suboptimality is incurred.

The estimation error does however lead to suboptimality, when the true and estimated values are in different segments. Then, the estimated value would yield HRP as the optimum policy, when HCP is the true optimum (ie, the optimum at the true value), or vice versa. Then, the amount of suboptimality is at most the difference in death counts attained by these two policies

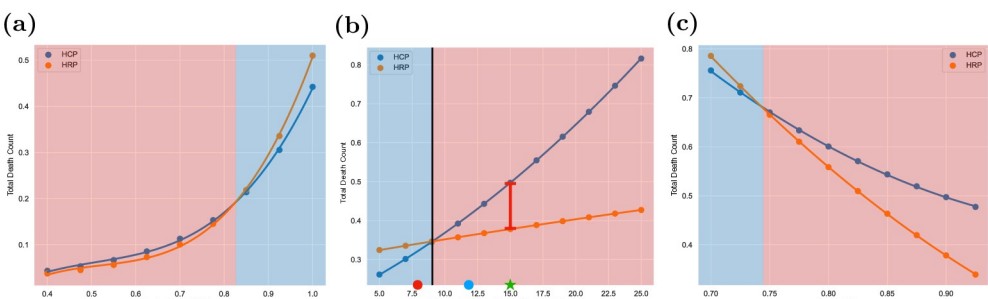

**Fig 14.** These figures show the death counts (as percent of the total population) of HCP (blue curve) and HRP (orange curve) vaccine policies as a function of the contact rate multiplier, high contact population size, and vaccine efficacy, respectively from left to right. The blue (respectively, red) shaded region represents the regime where HCP (respectively, HRP) is optimal. In Fig 14b, we identify the crossover threshold (where HCP regime switches to HRP or vice versa) by the vertical black line. As an example, suppose the true value of the high contact group size is 15% (indicated by the green star), but the estimated value is 11.5% (indicated by the blue circle on the x-axis). Then there is no suboptimality as both are in the region in which the same policy, namely HRP in this case, is deemed optimal. However, if the estimated value is 7.75% (indicated by the red circle on the x-axis), HCP, rather than HRP, would be computed optimal. The magnitude of the resulting suboptimality is indicated by the vertical red bar.

at the true value of the parameter (see illustration in Fig 14b). This difference is 0 when the true value of the parameter is the crossover point (that is, the point at which the HCP and HRP curves intersect) and increases as the distance between the true value and the crossover point increases. But if the true value of a parameter is far off from the crossover point, the estimated value can only be in a different segment (ie, on the other side of the crossover point) if the difference between the true and estimated values is large. This can only be caused by large values of the noise multiplier, which occur with low probability. Thus, the amount of suboptimality remains small with high probability. Thus, the system is robust to error in estimating these parameters.

The above explanations suggest that the suboptimality resulting from estimation error increases in: 1) the lengths of the segments in which the optimal policy is the same and the number of changes in the optimal policy 2) the difference in death counts attained by HCP and HRP. Fig 14 suggests that the first is similar for variation of all three parameters. But the second is different for variation of the three parameters. Specifically, the death counts of HCP and HRP are close for different values of the contact multiplier, while they differ most as high contact group size is varied. The difference is intermediate when vaccine efficacy is varied, but is closer to that for variation of the high contact group size. Thus, intuitively, we expect the suboptimality to be the smallest due to error in estimating contact rates, and larger for errors in estimating the high contact group size and vaccine efficacy. Fig 13 confirms this intuition. The suboptimalities incurred for noisy high contact group size and noisy vaccine efficacy are similar for low values of the variance $\sigma^2$ of the multiplicative noise distribution, but as this variance increases the suboptimality for the former is clearly larger. This is because for large variance, estimation errors are high with a higher probability, and when estimation errors are high, the true and estimated values can be in different segments even if the true value is far from the crossover point. When the true value of the high contact group size is far off from the crossover point, the difference in the death counts of HCP and HRP is larger, than when the true value of the vaccine efficacy is far off from the crossover point. This leads to higher expected suboptimality for errors in estimating the high contact group size than in the vaccine efficacy for high noise variance. Considering the expected suboptimality as a quantitative measure of robustness, we note that the system is least to most robust in errors in estimating the high contact population size, followed by vaccine efficacy and then contact rate multiplier.

When the true value of the vaccine efficacy is set at a value higher than 0.75 the suboptimality is much lower. This is because then the optimal policy is the same (specifically, HRP), for large ranges of the values of the other two parameters we vary.

Finally, note that Fig 14 also shows a sensitivity analysis of the optimal policy, which is geared towards further elucidating the robustness results. In each figure of Fig 14, we fix all but one parameter. This helps isolate the effects of specific parameters, and, thus, allows us to study robustness with respect to chosen parameters of interest, but at fixed values of other parameters. In contrast, for the sensitivity analysis done in Section 3.5, we vary all parameters in each figure, specifically for each value of one parameter, we studied a statistic for the aggregate of variations of all other parameters, which allowed us to analyze the sensitivity of the aggregate with respect to variation of the isolated parameter.

## 3.7 Different objective functions

We now consider optimal vaccination strategies that minimize the time averages of symptomatic and hospitalization counts as the public health objectives. The optimal control formulations that minimize these objectives can be found in S2 Appendix. Again we found that in most cases (90.91%) the optimal vaccination strategies was either the high contact or the high

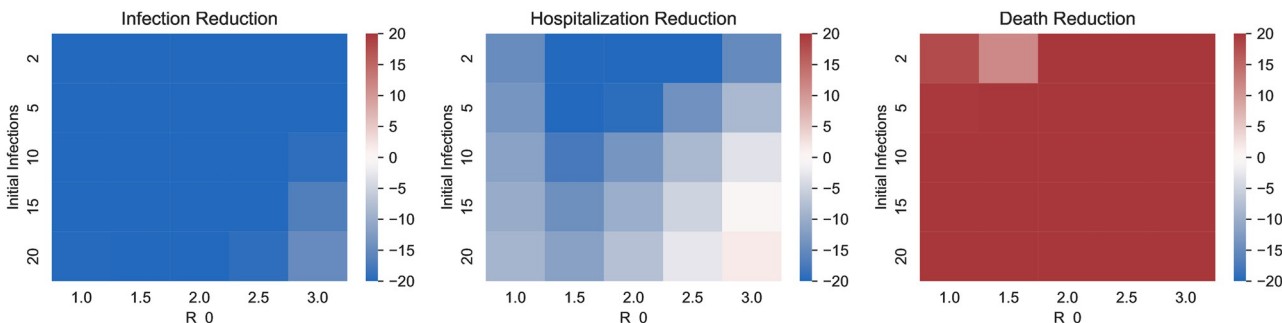

**Fig 15. Each heatmap displays the difference in resultant objective function between high contact prioritization and high risk prioritization vaccination strategies as a percentage of the objective value of high risk prioritization policy.** Fig 15a–15c respectively demonstrate the policy landscape when the objective is to minimize time average of symptomatic counts, time average of hospitalization counts, and overall death counts.

risk prioritization strategies. In the remaining cases, the average suboptimality of the better of the two was 2.26% and 5.99% for the two objectives respectively.

Fig 15 shows how the change in objective function affects the relative performances of the high contact and high risk prioritization strategies. The high contact prioritization strategy outperforms the high risk prioritization strategy for greater number of instances when minimizing symptomatics rather than deaths is the public health objective because the increased rates of hospitalization and death of the high risk individuals do not affect the symptomatic count. Similarly, minimizing hospitalizations reveals an intermediate result where high risk prioritization is more important than when minimizing symptomatics but less important than when minimizing deaths. This aligns with our intuition that the high risk group drives the largest share of the death count while the high contact group is predominantly responsible for spreading the disease.

### 3.8 Two dose vaccines

We formulate the problem of determining the optimal two dose vaccination strategy in S2 Appendix. When considering two dose vaccination, our decision space becomes richer as the second dose can be delayed by any given amount to accommodate the first doses for all or some groups. The decision problem therefore consists of the amount of delay for each group and the ordering of both doses. Different countries have implemented different delays and orderings for the two doses [67, 68]. There is an additional constraint now: the first dose must precede the second dose for every individual. In addition to the contributory factors considered so far, the optimal decision also depends on the degree of immunity provided by the first dose.

We ran our two dose model on the same parameter combination detailed in the first subsection. The range of single dose efficacy was 40% to 80% in intervals of 10% in line with clinical data from multiple available vaccines [69]. In the 4,556,250 instances we considered, the optimal policies were those that administer (1) both doses to high risk group, then both doses to high contact group, then both doses to the baseline group (60.91% of instances); (2) both doses to high contact group, then both doses to high risk group, then both doses to the baseline group (7.27% of instances); (3) first dose to high contact group, then first dose to high risk group, then second dose to high contact group, then second dose to high risk group, then both doses to baseline group (30% of instances); (4) both doses to high risk group, first dose to high contact group, first dose to baseline group, first dose to high contact group, then second dose to baseline group (1% of instances); and (5) first dose to high contact group, both doses to high

risk groups, second dose to high contact group, then both doses to baseline group (1% of instances). Clearly these candidate policies are easy to deploy. In 98% instances the optimal policy was one of the first three candidates. Overall, 69.18% of optimal policies (policies (1), (2), and (4)) did not delay second doses to accommodate first doses to multiple groups; in these *"non-delay"* cases the optimal strategy follows either the high contact or high risk prioritization policies and provides both doses to these entire groups in one go. In the remaining cases, the optimal policy opted to delay second doses for one or more groups. Thus, there are regimes where delaying second doses can provide significant additional mortality reduction.

Here it is useful to characterize when delaying second doses is beneficial in relation with our single dose studies. We will refer to instances where high risk prioritization was optimal in the single dose model as high risk prioritization regime and the same for high contact prioritization. We find that delaying the second dose is never beneficial in high risk prioritization regime. In this regime the optimal strategy focuses on those most vulnerable to the pandemic, therefore, interim efficacy after just one dose is not sufficient for this strategy. In high contact prioritization regimes, for sufficiently high interim vaccine efficacy ($>70\%$), we see that delaying the second dose of the high risk group and accommodating first doses to others within this delay can lead to significant mortality reduction (31.62% on average) over the best alternative non-delay policy. Put simply, if the protection of one dose is sufficiently high, then delaying second doses and accommodating first doses to others can help reduce mortality. It is also effective in reducing the number of infections and, thus, may effectively reduce the burden on health systems or economic impact of the pandemic in some regions.

## 3.9 Considering an additional overlap group

We have so far considered that the populace can be partitioned into three groups: (1) high contact (and low risk), (2) high risk (and low contact) (3) baseline (both low contact and low risk). This decomposition need not however cover the populace entirely. This is because in principle there can be an overlap between high contact and high risk groups, which this classification ignores. The rationale for ignoring this overlap is that it is minimal in practice, because the individuals in the high contact group are of working age and are usually in good health, therefore their risk of developing a severe form of the disease is generally low. The fact that this overlap is minimal throughout the world is objectively borne out from how closely our model based on the three group decomposition predicts the temporal variation of infection and death count which was observed in reality across the world (see section 3.1). It is difficult to reliably estimate the exact extent of this overlap even in US, but using demographic data from census, employment data from surveys, and health data for conditions that cause high risk from a variety of sources including CDC, we estimate that the overlap would be lower than 1.9% of the overall population in US—we believe the actual overlap to be significantly lower (refer to S6 Appendix for our calculations, and the reason why the actual overlap is likely to be significantly lower). Since however this overlap is positive, and can not be reliably estimated all over the world, we now augment our model with a fourth group which is comprised of overlap, that is, it is *both* high contact and high risk. This group, denoted as *W* or HCHR (high contact and high risk) henceforth, will have the contact rates of the high contact group *Z* (a.k.a. HCLR, high contact and low risk) and the symptomatic infection, hospitalization, and death probabilities of the high risk group *Y* (a.k.a. LCHR, low contact and high risk). In practice, *W* may refer to elderly high contact workers or high contact workers with comorbidities. In the new terminology, group *X*, which comprises of low contact and low risk individuals would be referred to as LCLR. The ODEs in Fig 2 are defined for general groups and can thus be applied directly now considering $i \in \{W, X, Y, Z\}$.

We numerically study this augmented model by solving instances using this model while varying contact rates, group sizes, COVID variants, $R_0$, NPI efficacy, and initial infection rate as in the results reported in Section 3.2. Since there are four groups to consider, we also additionally vary the size of group HCHR. Specifically, we consider HCHR group sizes of {0.1, 0.5, 1, 2, 5, 10}% of the total US population. We therefore run $\sim 5.5M$ instances, instead of $\sim 1$ M instances considered for the results reported in Section 3.2. Across all these instances, the HCHR group was *always* given first priority for vaccination by the optimal policy. Intuitively this is because individuals in this group spread the disease at the highest rate while they are in states from which the disease can be spread (that is the asymptomatic, presymptomatic, and early infection) and they have the highest risk of developing serious forms of the disease leading to hospitalization and death. Since usually the HCHR group is small in size the vaccination process for such individuals is concluded in a short time if they are accorded the first priority as the optimal policy does. Subsequently, the other groups ought to be vaccinated, whose relative priority in the optimal policy can be determined by solving the optimal control problem formulated for the three groups as in Section 2 and which has been reported in Sections 3.2 to 3.7. Thus for a wide range of parameters seen for COVID the optimal vaccination policy for four groups can be obtained through a simple and intuitive augmentation of that obtained for three groups.

## 4 Discussion

We now summarize how our contributions have advanced the state of the art on research on COVID-19 vaccination strategy. We have provided a framework for obtaining vaccination strategies that attain several desired public health objectives (e.g. minimizing symptomatic events, hospitalization, death counts) subject to vaccination capacity constraints. The framework incorporates both single dose and multidose vaccines, arbitrary combination of parameters and attributes that occur in practice such as reinfection, breakthrough infections. Our framework relies on the following methodological innovations: 1) capturing the innate heterogeneity of COVID-19 in contact and risk profiles through classification of the populace into three groups 2) formulating the optimization of a variety of public health objectives subject to vaccination capacity constraints as an optimal control problem with the state space evolution modeled as ODEs. We optimize vaccination strategy among literally all possible strategies including highly dynamic ones and without any constraint on how often the capacity allocations can be changed among groups. Such an optimal strategy, even when difficult to deploy, provides a valuable benchmark for comparisons of any proposed policy. Our work provides this benchmark, and goes beyond by providing a few (two to three depending on the number of doses of vaccines) easy-to-deploy strategies, at least one of which is optimal or near-optimal in a wide array of parameter combination. Our framework typically identifies optimal vaccination strategies within seconds for single dose vaccine using modest computation resources which are usually available in public health centers including in the LMICs. For multi-dose vaccines the computation time merely increases to minutes. Owing to this computational tractability we could present results for large landscapes of 911,250 instances involving variations of parameters in large realistic ranges (Section 3). Large landscapes allow for more reliable conclusions on how the optimal strategy and the optimal value of the public health objective changes with variations in parameter values; large landscapes also help identify near-optimal, easy-to-deploy vaccination strategies which we accomplish. In contrast, even for the simplest scenario of single dose vaccines, no breakthrough or reinfection, the previous works are largely computationally intensive. Most of the previous published work has been limited to evaluations and comparisons of a handful of vaccination strategies by simulations; a smaller class of

previous work optimized among a restricted class of strategies which either do not change capacity allocation to groups at all (i.e. fully static strategies) or change once in a large predetermined decision intervals (i.e. limited dynamism). We show that allowing fully dynamic allocation of vaccine capacities among different population groups (i.e. not restricting the size of decision intervals) substantially enhances public health metrics; the performance in fact improves with increase in the allowed dynamism. Our vaccination strategy is therefore able to significantly outperform the static or limited-dynamism strategies. Our framework also enables the analysis of sensitivity of the optimal vaccination strategy and the optimum value of public health objective functions to variation of crucial parameters, which we do for realistic ranges of a variety of parameters with minimizing death count as the public health objective.

Our framework does assume the knowledge of relevant disease parameters, namely the contact rates of different groups, size of groups, vaccine efficacy, and disease parameters (transmissibility and duration of different disease states and rates of symptomatic infection, hospitalization, and death). Our robustness study (see Section 3.6) shows that the optimal policies are generally robust to imperfect knowledge of disease parameters, group sizes, vaccine efficacy, and contact rates. Focusing on contact rates, vaccine efficacy and group sizes, we explain why this is the case and identify key attributes which determine the extent of this robustness. Considering quantitative criteria for assessing robustness, we find that the robustness is the highest (respectively, lowest) to errors in estimating contact rates (respectively, size of the high contact group). The disease parameters are also consistent over time for a given strain of virus. The contact rates, however, may well change over the course of the pandemic even for a single strain. As detailed in Section 3.1, we allow contact rates to change every two months to account for infrequent shifts (i.e. lockdown changes and school openings); however, the controller is assumed to have knowledge of these contact rates over the entire course of the pandemic.

COVID-19 is not over yet and new variants are emerging in different parts of the world. It is unclear if the same vaccines will be effective for future variants. If not, then the vaccination process will need to restart from scratch. In addition, given how frequently pandemics have recurred in recorded human history [70], an entirely new pandemic is likely to emerge at some point in future, and it is imperative to be prepared for countering the same with a well-founded vaccination strategy. We describe how this paper can provide a vaccination strategy for a future variant or pandemic.

The stages of the disease we consider appear with some variation in different contagious diseases which spread through contact. The variations are likely to be in the transition rates between the stages and in risk profiles. The distinguishing aspect of our model is that the risks of hospitalization and death significantly increase with age and poor health. Our framework will provide the optimal, or near-optimal vaccination strategy, whenever these general principles hold. This is likely to be the case for future variants and possibly for one or more future pandemics. In such instances, the framework will proceed as follows:

1. Following the classification strategy we proposed for COVID-19 (Section 2), divide the populace into the same three groups: 1) high risk (e.g. retirees) 2) high contact (those with large and dynamic contact sets owing to professional requirements) 3) baseline (rest).

2. For single dose vaccine, compare the value of the objective function of interest (death, hospitalization, symptomatic counts) for high contact and high risk priority vaccination strategies. Deploy the better of the two.

3. For two dose vaccines, compare as above the three candidate optimal vaccination strategies of Section 3.8 and deploy the best. The comparisons can be performed using the heat maps

of Figs 6, 7 and 15 or from the ODEs of Fig 2 and the ODEs associated with the generalized model (see S2 Appendix).

For possible future pandemics, and current evolving pandemic response strategies, our framework's flexibility, computational efficiency, and usability present a new practicable pathway for systematically reasoning about optimal vaccination policies.

## Supporting information

**S1 Appendix. Symbol tables and values.**
(PDF)

**S2 Appendix. Generalizations of the basic model.**
(PDF)

**S3 Appendix. Proof of Theorem 4.1.**
(PDF)

**S4 Appendix. Numerical evaluations.**
(PDF)

**S5 Appendix. Runtime performance and comparison.**
(PDF)

**S6 Appendix. Estimation of the size of the overlap group.**
(PDF)

## Author Contributions

**Conceptualization:** Raghu Arghal, Harvey Rubin, Shirin Saeedi Bidokhti, Saswati Sarkar.

**Data curation:** Raghu Arghal.

**Formal analysis:** Raghu Arghal, Shirin Saeedi Bidokhti.

**Investigation:** Raghu Arghal, Harvey Rubin, Shirin Saeedi Bidokhti, Saswati Sarkar.

**Methodology:** Raghu Arghal, Harvey Rubin, Shirin Saeedi Bidokhti, Saswati Sarkar.

**Project administration:** Shirin Saeedi Bidokhti, Saswati Sarkar.

**Software:** Raghu Arghal.

**Supervision:** Harvey Rubin, Shirin Saeedi Bidokhti, Saswati Sarkar.

**Validation:** Raghu Arghal.

**Visualization:** Raghu Arghal, Saswati Sarkar.

**Writing – original draft:** Raghu Arghal, Saswati Sarkar.

**Writing – review & editing:** Raghu Arghal, Harvey Rubin, Shirin Saeedi Bidokhti, Saswati Sarkar.

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
