## [Decision Letter · Decision Letter 0]

20 Sep 2024

PONE-D-24-21798Protect or prevent? A practicable framework for the dilemmas of COVID-19 vaccine prioritizationPLOS ONE

Dear Dr. Arghal,

Thank you for submitting your manuscript to PLOS ONE. Firstly, we would like to apologize for the delay in processing your manuscript. It has been exceptionally difficult to secure reviewers to evaluate your study. We have now received one completed review, which is available below. The reviewer has raised significant scientific concerns about the study that need to be addressed in a revision.

Please note that we have only been able to secure a single reviewer to assess your manuscript. We are issuing a decision on your manuscript at this point to prevent further delays in the evaluation of your manuscript. Please be aware that the editor who handles your revised manuscript might find it necessary to invite additional reviewers to assess this work once the revised manuscript is submitted. However, we will aim to proceed on the basis of this single review if possible. 

We look forward to receiving your revised manuscript.

Kind regards,

Miquel Vall-llosera Camps

Senior Staff Editor

PLOS ONE

Journal requirements: 1. When submitting your revision, we need you to address these additional requirements. Please ensure that your manuscript meets PLOS ONE's style requirements, including those for file naming. The PLOS ONE style templates can be found at https://journals.plos.org/plosone/s/file?id=wjVg/PLOSOne_formatting_sample_main_body.pdf and https://journals.plos.org/plosone/s/file?id=ba62/PLOSOne_formatting_sample_title_authors_affiliations.pdf. 2. Please provide a complete Data Availability Statement in the submission form, ensuring you include all necessary access information or a reason for why you are unable to make your data freely accessible. If your research concerns only data provided within your submission, please write "All data are in the manuscript and/or supporting information files" as your Data Availability Statement. 3. Please include captions for your Supporting Information files at the end of your manuscript, and update any in-text citations to match accordingly. Please see our Supporting Information guidelines for more information: http://journals.plos.org/plosone/s/supporting-information. 

Reviewers' comments:

Reviewer's Responses to Questions

**Comments to the Author**

1. Is the manuscript technically sound, and do the data support the conclusions?

Reviewer #1: Yes

2. Has the statistical analysis been performed appropriately and rigorously? 

Reviewer #1: Yes

3. Have the authors made all data underlying the findings in their manuscript fully available?

Reviewer #1: Yes

4. Is the manuscript presented in an intelligible fashion and written in standard English?

Reviewer #1: Yes

5. Review Comments to the Author

Reviewer #1: Suggestions for minor revision:

1. Authors should expand the Discussion on Group Overlaps: Provide a more detailed discussion on the potential overlap between high contact and high risk groups, including any assumptions or data supporting the claim that this overlap is minimal.

2. Authors should include Empirical Data or References: Strengthen the rationale for group-specific contact rates and risk factors with empirical data or references to existing studies.

3. Authors should incorporate Sensitivity Analysis: Add a section on sensitivity analysis to explore how different parameter values might affect the model outcomes.

4. Authors should explanation of Numerical Methods: The methodology mentions using numerical toolboxes Yop and CasADi for solving the optimal control problem. However, it lacks a detailed description of how these tools are used, including any specific algorithms or settings. Providing this detail would make the methodology more reproducible.

5. To better understand the robustness of the model's outcomes, i recommened, if posssible, the authors include a sensitivity analysis to explore how changes in key parameters (e.g., contact rates, vaccine efficacy, or group sizes) might affect results. This would help determine the model's reliability under different scenarios.

6. PLOS authors have the option to publish the peer review history of their article (what does this mean?). If published, this will include your full peer review and any attached files.

Reviewer #1: No

---

## [Author Response · Author response to Decision Letter 0]

13 Nov 2024

Our response to the reviewer is attached as a file in the submission.

---

## [Editor Report · Decision Letter 1]

10 Dec 2024

Protect or prevent? A practicable framework for the dilemmas of COVID-19 vaccine prioritization

PONE-D-24-21798R1

Dear Dr. Arghal,

We’re pleased to inform you that your manuscript has been judged scientifically suitable for publication and will be formally accepted for publication once it meets all outstanding technical requirements.

Kind regards,

Osmond Ekwebelem

Academic Editor

PLOS ONE
---

## [Editor Report · Acceptance letter]

18 Dec 2024

PONE-D-24-21798R1 

PLOS ONE

Dear Dr. Arghal, 

I'm pleased to inform you that your manuscript has been deemed suitable for publication in PLOS ONE. Congratulations! Your manuscript is now being handed over to our production team.

Kind regards, 

on behalf of

Dr. Osmond Ekwebelem 

Academic Editor

PLOS ONE